# Scaling Channel-Adaptive Self-Supervised Learning

**Alice V. De Lorenci[1], Seungeun Yi[2], Theo Moutakanni[2], Piotr Bojanowski[2], Camille Couprie[2]\*, Juan C. Caicedo[3], and Wolfgang M. Pernice[4]\***

[1] *University of São Paulo, Brasil (Work done during an internship at FAIR, Meta.)*,
[2] *Meta FAIR Paris, France*, [3] *University of Wisconsin–Madison, USA*, [4] *Columbia University, New York, USA*
*\* Corresponding Authors*

*Reviewed on OpenReview:* https://openreview.net/forum?id=pT8sgtRVAf

## Abstract

Recent advances in self-supervised pre-training of foundation models for natural images have made them a popular choice for various visual systems and applications. Self-supervised strategies are also promising in non-RGB scientific imaging domains such as in biology, medical and satellite imagery, but their broader application is hampered by heterogeneity in channel composition and semantics between relevant datasets: two datasets may contain different numbers of channels, and these may reveal distinct aspects of an object or scene. Recent works on channel adaptive strategies report substantial advantages for those that account for variable channel compositions without sacrificing the ability to jointly encode channels; yet, how these strategies behave at scale remains unclear. We here show that, surprisingly, trained across large-scale datasets, independent-encoding of channels outperforms joint-encoding methods by a substantial margin. We validate this result along an extensive set of experiments on various datasets from cell microscopy to geospatial imagery. Our DINO BoC approach sets a new state-of-the-art across challenging benchmarks, including generalization to out-of-distribution tasks and unseen channel combinations. We open-source code and model weights for a new general-purpose feature extractor for fluorescent microscopy.

## 1 Introduction

Advances in self-supervised learning (SSL) recently enabled pretraining of powerful general feature extractors across large collections of unlabeled natural imaging data (Oquab et al., 2023; He et al., 2022; Chen et al., 2020). This approach also holds major promise for scientific imaging domains. Combined with ever more powerful methods to automate image-acquisition across thousands of single cells and conditions, foundation models for fluorecent microscopy stand to substantially empower biologists to query mechanisms of disease, to accelerate drug discovery, and to interrogate cellular biology at unprecedented scale and salient detail (Chandrasekaran et al., 2021). Earth observation foundation models could help address important societal challenges, such as by estimating aboveground biomass (Muszynski et al., 2024), tree canopy heights (Tolan et al., 2024), classifying methane sources (Zhu et al., 2022), and beyond (Lacoste et al., 2023). Yet, the application of SSL methods in these and related domains face a key

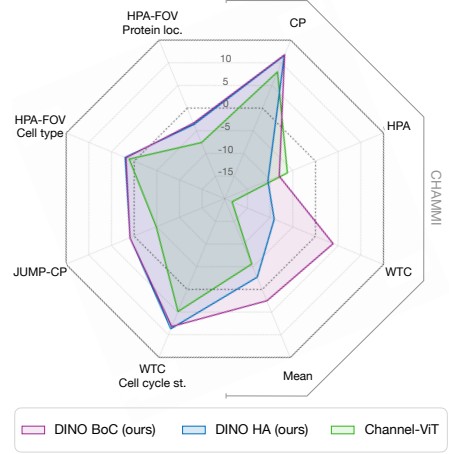

Figure 1: DINO-BoC is a channel-agnostic approach for SSL in channel-heterogeneous data domains that outperforms channel-adaptive methods and advances the SOTA across a variety of benchmarks in cellular microscopy (results relative to specialized baselines).

technical challenge: scientific images often feature diverse sets of channels, with variable semantics, tailored to address specific questions. For example, to observe a given biological phenomenon of interest, biologists routinely design bespoke imaging protocols to reveal specific sets of cellular structures to the exclusion of others. The data-landscape of fluorescent microscopy thus consists of a vast number of small-to-moderate scale datasets with heterogeneous channel combinations (see Fig. 2A). In contrast to natural (RGB) images, most models trained on one fluorescent microscopy dataset can thus neither be re-used in other studies, nor draw on other datasets, yielding representations of limited expressivity and that generalize poorly (Chen et al., 2024). Similar issues arise in geospatial data analysis (Fig. 2B). Learning powerful unified feature extraction models (i.e. foundation models) for scientific imaging domains therefore requires methods that accommodate heterogeneous channel combinations which we describe broadly as channel-adaptive models.

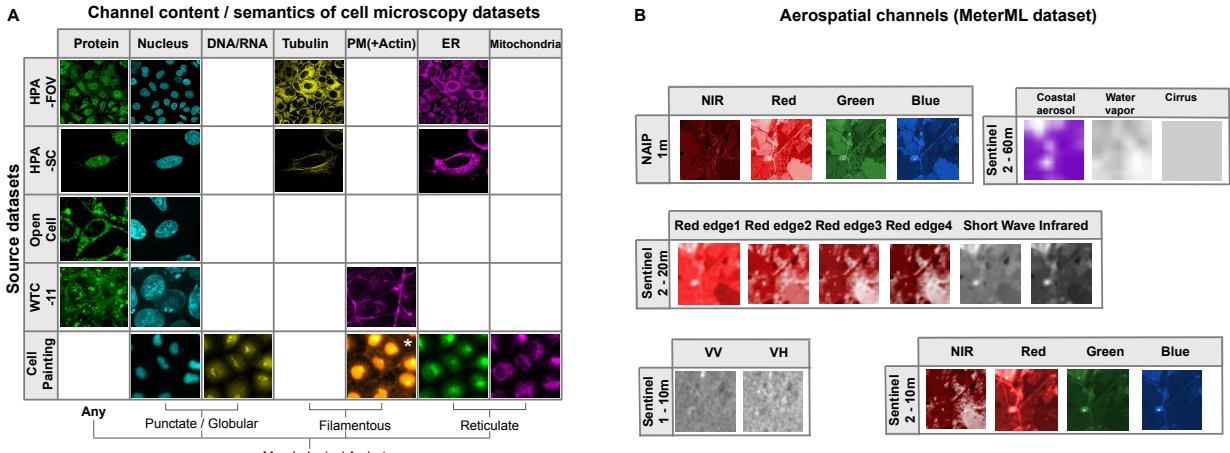

Figure 2: **Training across datasets with various numbers of channels.** (A) We consolidate images from Human Protein Atlas (2019)(HPA-FOV), Le et al. (2022)(HPA-SC), Cho et al. (2022)(OpenCell), Viana et al. (2023)(WTC) and Doron et al. (2023)(Cell Painting), into a dataset (ExtendedCHAMMI) that reflects the diversity in channel number, order, and semantics, that characterizes fluorescent microscopy studies. Despite some conventions (e.g. the nucleus is usually imaged using blue fluorescing stains), there is no necessary correspondence between specific channels and/or wavelengths, and their biological semantics. Images are pseudo-colored according to emission wavelength. CellPainting visualizes Plasma Membrane (PM) and Actin in one channel. (B) In aero-spatial imagery, images acquired at different resolution and using different wavelength / acquisition protocols are often available, but it remains a challenge to fully exploit all image modalities. We benchmark models on the MeterML dataset (Zhu et al., 2022), one sample is shown here.

Fluorescent microscopy has proven a particularly fruitful test-bed for the development of such methods. As proposed by Xun et al. (2024), a technically simple solution to the challenge of variable channel inputs is to model channels independently by passing them through the model one-by-one, and by reassembling full-image representations by concatenating channel-specific output embeddings post-hoc. This has the theoretical advantage of yielding fully *channel-agnostic* models that are inherently compatible with arbitrary channel sets, but sacrifices the ability to (explicitly) learn channel-interactions. This strategy noticeably breaks with established best practices for RGB images, and one may expect channel-agnostic models to broadly fail in key analytical use cases that require cross-channel reasoning, such as the analysis of protein-localization relative to reference channels that provide ground-truth on organelle position (Human Protein Atlas, 2019; Lacoste et al., 2024). Consistently, recent works report significant advantages for more complex approaches that reconcile conventional joint-channel-encoding with variable number of input channels (up to some maximum) through customizations of vision transformers (ViTs) (Bao et al., 2024; Bourriez et al., 2024; Pham & Plummer, 2024). Yet, previous studies employed inconsistent sets of comparatively small pre-training datasets, often with limited channel-diversity, making their results difficult to compare. They further relied either only on supervised learning objectives (Bao et al., 2024), and/or did not evaluate generalization to out-of-distribution

(OOD) tasks and datasets with unseen channel combinations - key metrics of success that will determine the utility of general purpose channel-adaptive feature extractors in practice.

We here conduct a first, large-scale comparative study into the scaling properties of channel-adaptive methods, and compare their performance across uniform model architectures, learning objectives, and rigorous benchmarks. We note that inconsistent and often missing labels across microscopy (and other scientific) datasets render supervised methods ill-suited to the task of learning foundation models. We hence focus our work on state-of-the-art (SOTA) SSL strategies which have been recently shown to yield rich representations of cellular morphology on channel-homogeneous microscopy datasets (Doron et al., 2023; Kraus et al., 2024).

Our study finds that, at scale, self-supervised ViT models (Oquab et al., 2023) trained on individual channels via a Bag of Channels (BoC) approach significantly and outperform models that jointly encode channels across an extensive set of testing regimes in microscopy as well as geospatial data. Our results contradict previous reports and challenge the assumption that joint-channel-encoding is beneficial in non-RGB domains. Our main contributions are as follows:

- We conduct a systematic comparative study of channel-adaptive SSL methods across large and diverse datasets.

- We report that, surprisingly, independent-encoding of channels outperforms joint-channel-encoding strategies across an extensive set of experiments including in-domain, cross-dataset, and OOD generalization setups, challenging key assumptions on the optimality of joint-channel-encoding in non-RGB domains.

- We substantiate our results through a set of control experiments to directly analyze the impact of joint versus independent-channel-encoding by virtue of a novel channel-adaptive Hierarchical Attention scheme, as well as an ablation of SSL objectives.

- We open-source a new general-purpose feature extractor, DINO-BoC, that sets a new SOTA for cross-dataset learning in fluorescent microscopy and is compatible with arbitrary channel combinations. Our approach also achieves SOTA results on the geospatial Meter-ML dataset when using all channels.

## 2 Related Work

**Self-supervised learning.** SSL describes a family of methods that leverage general pretext tasks to learn rich and ever more complete representations of the relationships inherent to a data domain directly from its samples, without the need for supervisory labels. The success of SSL both in Natural Language Processing (Devlin et al., 2019; Radford et al., 2019; Touvron et al., 2023) and, more recently, in Computer Vision (Chen et al., 2020; Caron et al., 2021; He et al., 2022; Assran et al., 2023) can in large part be attributed to well-documented – if still only incompletely understood – scaling properties that in particular transformer-based architectures trained with SSL methods exhibit with growing dataset and model sizes. SSL methods in vision can be broadly divided into generative strategies, such as Masked Autoencoders (MAE) (He et al., 2022), which are trained by reconstructing masked or corrupted regions of the input, and contrastive methods, including DINOv2 (Oquab et al., 2023), the current state-of-the-art (SOTA) method for large-scale SSL for natural images. We investigate the extent to which the scaling properties of SSL methods generalize to scientific imaging domains with heterogeneous channel composition.

**Fixed Channel SSL in fluorescent microscopy.** Both contrastive and generative SSL approaches have been recently shown to hold promise for the derivation of powerful, unbiased, general-purpose feature extractors for scientific imaging domains, including cellular microscopy (Doron et al., 2023; Kobayashi et al., 2022). However, unable to accommodate heterogeneous channel combinations, these studies remained confined to datasets-specific models with limited scale. Our study pioneers large-scale cross-dataset SSL for domains with variable channel inputs and delivers a new SOTA general-purpose feature extractor for this domain.

**SSL for geospatial imaging.** Similar to fluorescent microscopy, the majority of existing "foundation models" for geospatial imaging data (Szwarcman et al., 2024; Reed et al., 2023; Cong et al., 2022) are trained on fixed sets of channels (bands). This limits their broader application across channel-heterogeneous

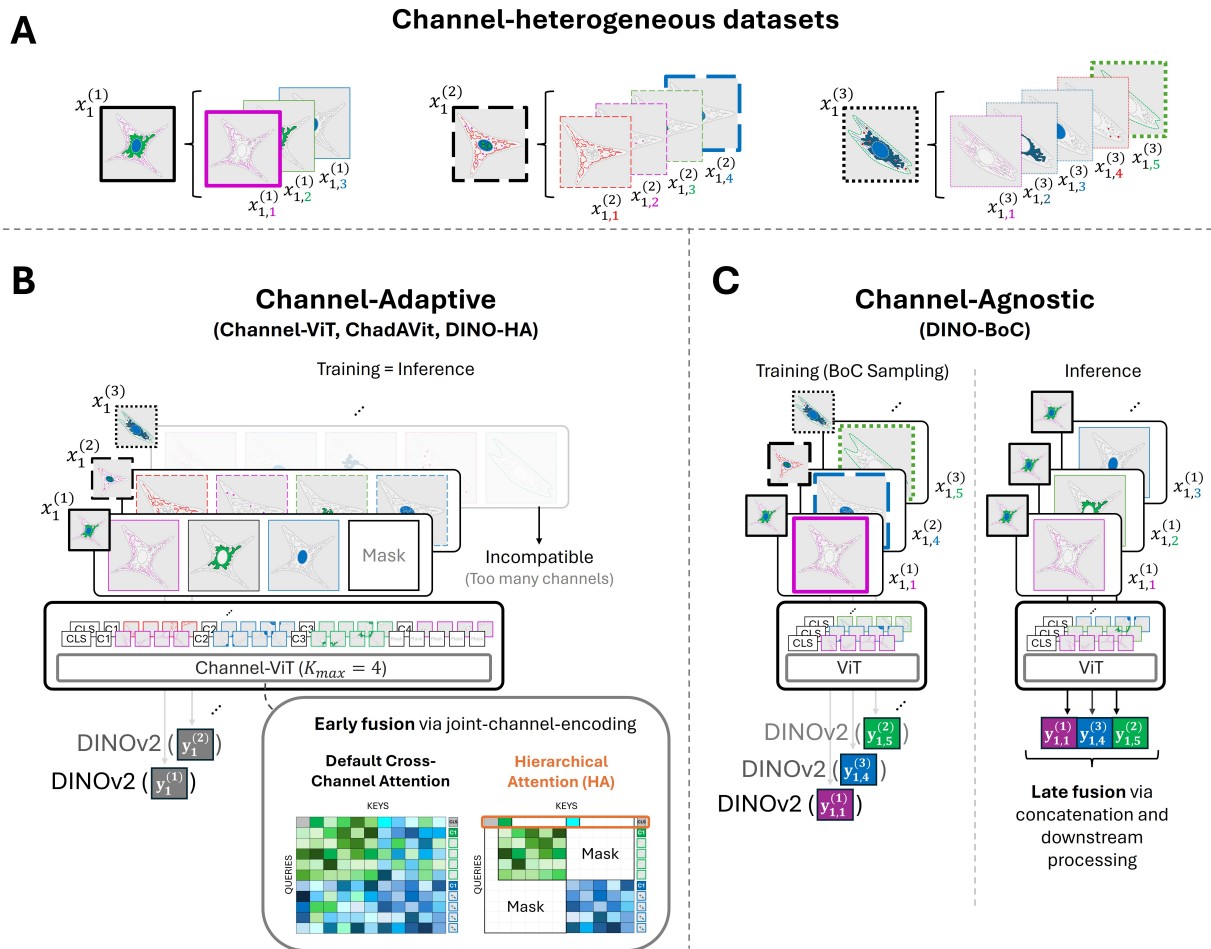

Figure 3: Strategies for datasets with (A) variable-channel compositions (solid, dashed, dotted borders). (B) Channel-Adaptive methods, once trained, can accommodate variable channel numbers up to a maximum (here $K_{\max}$=4), with full cross-channel computation (akin to early fusion). *Inset:* Our Hierarchical Attention approach limits cross-channel computation for channel-adaptive methods to the class token. (C) DINO-BoC is similar to late-fusion multi-modal strategies and trains on individual channels, randomly drawn (thick borders in A) from the datasets (no cross-channel computation). On inference, the encoder can be used to embed images with arbitrary channel combinations by concatenating channel-wise outputs.

datasets. Moreover, most of these earth observation foundation models use MAE as pre-training strategy. Yet, comparisons on satellite benchmarks (Wang et al., 2022) exhibit superior performance of DINO over MAE. Anysat (Astruc et al., 2024) is the first approach to allow for pretraining on datasets with an heterogeneous number of channels in geospatial imaging. However, this approach relies on domain-specific metadata for positional embedding (based on ground sample distance) that are unavailable in other scientific domains. In contrast, our approach is generic and compatible with any imaging domain.

**Channel-Adaptive SSL.** To account for variable channel inputs while allowing for cross-channel computation, Channel-ViT (Bao et al., 2024), ChAda-ViT (Bourriez et al., 2024), DiChaViT (Pham & Plummer, 2024), and Kraus et al. (2024) convert the variable-channels problem into one of variable sequence length, which transformers natively support. Kraus et al. (2024) uses MAE with channel-specific decoders during training, while Channel-ViT, ChAda-ViT, and DiChaViT introduce learnable channel-embeddings to preserve channel-specific information. (Bao et al., 2024) uses supervised learning objectives, and additionally proposes a hierarchical channel sampling (HCS) approach during training. (Bourriez et al., 2024) trains with DINO

(Caron et al., 2021) with a ViT-S on relatively small datasets. We uniformly use ViT-L architectures, and systematically compare the scaling properties of channel-adaptive joint-channel-encoding to channel-independent encoding across large and diverse pre-taining datasets and extensive benchmarks.

In particular, an alternative approach for cross-dataset learning in variable-channel domains is to ignore the association between channels altogether, by training on single-channel (i.e. gray-scale) inputs, yielding *Channel-Agnostic* models. CytoImageNet (Hua et al., 2021) proposed to collapse channels into one by averaging across them. This approach irrevocably discards the semantic information carried by distinct channels. Closest to DINO-BoC, Microsnoop (Xun et al., 2024) instead trains on individual channels sampled from a small custom dataset (about 10k images) that includes fluorescent microscopy images, using a convolutional U-Net (Ronneberger et al., 2015) with a masked SSL strategy. Whole-image representations are reassembled post-hoc by concatenating output embeddings for each channel. However, pre-training data and training code are not available. We here study BoC sampling for large-scale channel-heterogeneous cross-dataset training, using ViTs and SOTA SSL objectives, in systematic comparison to contemporary channel-adaptive methods, and two distinct scientific imaging domains. DINO-BoC outperforms Microsnoop by a large margin and is fully open-source. While different in approach, we here conceptually treat BoC models as a special case of Channel-Adaptive approaches. We further explore the continuum between models with full vs. no cross-channel computation by virtue of a new Channel-Adaptive ViT architecture with a *Hierarchical Attention* (HA) mechanism (see Fig. 3).

**Multi-modal learning.** Channel-adaptive learning shares technical similarities with multi-modal learning if one were to interpret channels as different data modalities. For example, MultiMAE (Bachmann et al., 2022) extends the MAE pre-training objective to handle a variable number of modalities by randomly sampling patches from distinct modalities that are passed through a common transformer encoder. The training objective is to reconstruct the masked-out patches using task specific decoders. By building the input sequence with patches from different modalities, MultiMAE resembles joint-channel-encoding strategies. Contrastive learning has also been explored in the context of multi-modality (Liang et al., 2023). Here, however, the different views are built from different modalities, whereas in case of channel-adaptive models, views are constructed through image augmentations (e.g. random cropping), such that each view contains the same channels. (Madaan et al., 2024) consider the dichotomy and possible tradeoff between capturing inter- vs. intra-modality dependencies, albeit in a supervised learning scenario. In contrast to channel-adaptive strategies that encode channels jointly, BoC models cannot explicitly learn inter-channel relationships and hence allocate all capacity to intra-channel dependencies. Finally, the multi-modal literature distinguishes approaches performing early fusion of features – at the first layers of models – or late fusion, i.e. feature combination at the last layers. Analogously, joint-channel-encoding strategies perform early fusion, whereas the BoC approach resembles late fusion methods (Fig. 3).

## 3 Method

Given a channel-heterogeneous dataset $\mathbb{X}$, where a sample $x^{(j)}$ has $K_j$ channels, let $K_{\max}$ denote the maximum number of channels across all images. The goal of channel-adaptive models is to accommodate images with a heterogeneous channel combinations.

### 3.1 Channel-Adaptive SSL baseline: Channel ViT

We implement joint-channel-encoding strategies following Channel-ViT (Bao et al., 2024), ChAda-ViT (Bourriez et al., 2024), and DiChaViT (Pham & Plummer, 2024), that adapt the patchfication process of standard ViTs and add channel embeddings (see Fig. 3B, Fig. 3A). Specifically, given an input image with $K$ channels $x \in \mathbb{R}^{K \times H \times W}$, where $(H, W)$ is the resolution of the original image, and a patch size $S$, we reshape an image into a sequence of flattened single-channel patches of length $N = KHW/S^2$:

$$x = \begin{bmatrix} x_{1,1} & \cdots & x_{1,N} & \cdots & x_{K,1} & \cdots & x_{K,N} \end{bmatrix}, \quad x_{k,i} \in \mathbb{R}^{S^2},$$

where, in $x_{k,i}$, $k$ indicates which channel the patch belongs to and $i$ its raster position. A linear embedding layer is applied to each patch $x_{k,i}$ resulting in a sequence of patch embeddings $h_i$ of dimension $D$, to which a learnable class token $x_{\text{CLS}}$ is prepended. To retain channel information, learnable channel embeddings are added along with the usual positional embedding to the patch embeddings $h_{k,i}$. Let $p_i$ $(i = 1, \ldots, N)$ denote the positional embeddings and $c_k$ $(k = 1, \ldots, K)$ the channel embeddings. The resulting sequence of patch embeddings for a sample $x$ is:

$$\begin{bmatrix} x_{\text{CLS}} & h_{1,1} + p_1 + c_1 & \cdots & h_{K,N} + p_N + c_K \end{bmatrix} \in \mathbb{R}^{D \times (1+NK)}.$$

In result, the variable number of channels problem turns into one of variable sequence length, benefiting from the transformer's inherent capability of handling arbitrary sequence lengths. Note that if the maximum number of channels per image on the pre-training data is $K_{\max}$, this method cannot be used on images that have more channels than $K_{\max}$, as there will be no trained channel embeddings for the extra channels. We scale this approach using a ViT-L architecture and update the pre-training recipe to the improved DINOv2. For terminological convenience we refer to these model configurations as *Channel-ViTs* (Fig. 3B).

## 3.2 Channel-Agnostic SSL: DINO-BoC.

A straightforward strategy to deal with channel number variability is to separately encode each channel, using a common backbone (Figure 3C); we will refer to this strategy as the *Bag of Channels* approach. During training, each input to the model consists of a random single channel extracted from a random multi-channel image, reshaped into a flattened of sequence of length $N = HW/S^2$:

$$x = \begin{bmatrix} x_1 & \cdots & x_N \end{bmatrix}, \quad x_i \in \mathbb{R}^{S^2},$$

After projecting the patches with a linear embedding layer, we add positional embeddings and prepend the class tokens to obtain for a sample $x$:

$$\begin{bmatrix} x_{\text{CLS}} & h_1 + p_1 & \cdots & h_N + p_N \end{bmatrix} \in \mathbb{R}^{D \times (1+N)}$$

At inference time, features are independently extracted for each channel of the image using the shared backbone. These channel-wise features are then aggregated – e.g., via concatenation – to obtain an image-level representation.

## 3.3 Channel Adaptive Hierarchical Attention model

This work introduces a novel baseline that balances joint and independent-channel-encoding strategies. This method can be seen as exploring the trade-off between Channel-ViT and DINO BoC, by limiting inter-channel interactions.

In this approach, the image is reshaped into a sequence of single-channel patches $x \in \mathbb{R}^{S^2 \times NK}$. After embedding the patch tokens with a linear layer, both a global class token $x_{\text{CLS}}$ and channel class tokens $x_{\text{CH}}$ are inserted into the sequence, only position embeddings are used:

$$\begin{bmatrix} x_{\text{CLS}} & x_{\text{CH}_1} & h_{1,1} + p_1 & \cdots & h_{1,N} + p_N & \cdots & x_{\text{CH}_\text{K}} & h_{K,1} + p_1 & \cdots & h_{K,N} + p_N \end{bmatrix}.$$

A specialized attention mask is employed in the Multi-Head Self-Attention blocks. Tokens within a single channel (channel class token and corresponding patch tokens) can only attend to other tokens within that same channel, therefore, at this level, the channels are processed independently. The global class token, however, attends to all channel class tokens, enabling inter-channel reasoning at a higher semantic level. This approach, termed DINO *Hierarchical Attention* (DINO HA) model, and the attention mask are illustrated in Figure 4 B,D (see Appendix D for more details).

## 4 Experiments

We test the merits of DINO BoC on diverse benchmarks, and compare it to existing channel-adaptive strategies. After introducing datasets and implementation details in Section 4.1, Section 4.2 demonstrates the impact of choosing the SSL DINOv2 method (Oquab et al., 2023) instead of MAE (He et al., 2022), and compares DINO BoC to Channel-ViT and Microsnoop (Xun et al., 2024). In section 4.3, we further investigate the advantages of DINO BoC on cross-dataset generalization tasks. Then, in section 4.4 we evaluate self-supervised DINO BoC, Channel-ViT, and DINO HA models in the CHAMMI benchmark – designed to assess performance in in-distribution and OOD tasks – compared to state-of-the-art supervised channel-adaptive approaches. Section 4.5 compares results achieved on an aero-spatial dataset. Finally, Section 4.6 assesses the impact of the BoC strategy with respect to usual training on RGB images.

### 4.1 Datasets, benchmarks and implementation details

#### 4.1.1 Datasets

We leverage multiple microscopy datasets with varying numbers of channels. In particular, we use the Human Protein Atlas, WTC-11, JUMP-CP and Cyclops datasets for evaluation tasks. Additionally, we employ the CHAMMI benchmark, a standardized evaluation framework for channel-adaptive models.

**Human Protein Atlas dataset.** The subset of the Human Protein Atlas (HPA) data that is considered is the one of the Kaggle competition Human Protein Atlas (2019), concerned with the subcellular distribution of the proteins encoded by different genes. It covers 35 cell lines and 28 subcellular structures of protein localization. There are $113,545$ images in total, with four channels. There is also a single cell version of the same dataset (Le et al., 2022), obtained through segmentation of the field-of-view (FOV) images. The *HPA Single Cell* dataset contains $839,612$ images.

**WTC-11 dataset.** This dataset is a version of the WTC-11 hiPSC Single-Cell Image Dataset v1 (Viana et al., 2023) of the Allen Institute curated for the CytoData Symposium 2022 hackathon (Allen Institute, 2022). The dataset contains $214,037$ 3D images of cells, we used the maximum z-projection of the original images. The dataset provides cell-cycle stage annotations, with six stages. The images have one bright-field (BF) channel and three fluorescence channels; BF was excluded.

**Cell Painting dataset.** We utilize the Cell Painting (CP) dataset curated by Moshkov et al. (2024), totaling $8,423,455$ images with five channels. The dataset has the objective of allowing the study of the response of cells to different compound treatments and gene over-expression experiments.

**JUMP-CP dataset.** This dataset, used by Bao et al. (2024), is a processed version of the data made available by the JUMP-Cell Painting Consortium (Broad Institute, 2021). It contains $229,228$ single cell images. We used only the five fluorescence channels in our work.

**Cyclops dataset.** We used the same dataset as described by Xun et al. (2024) consisting of 28,166 2-channel yeast cell images from the Cyclops database (Lu et al., 2018).

**OpenCell dataset.** The OpenCell dataset was introduced by Kobayashi et al. (2022), and encompasses $1,311$ different tagged proteins. In total, $1,134,592$ images were made available, with two fluorescence channels. More details on the five datasets mentioned above are given in Appendix B.

**CHAMMI benchmark.** The CHAMMI benchmark (Chen et al., 2024) includes a dataset curated from the WTC-11, HPA Single Cell and Cell Painting datasets. In total there are $220,284$ images, of which $100,145$ are used for training. The benchmark is a standardized evaluation framework for channel-adaptive models. It presents a comprehensive set of nine tasks for channel-adaptive models of varying complexity, that evaluate the ability of the models to generalize to new biologically-relevant experimental regimes. As such it positions itself as an indispensable benchmark to evaluate those models. The images from each data source present in the CHAMMI dataset are split into one training set and several test sets, designed for specific tasks. Tasks with suffix 1 are IID classification problems, where the test and train data follow the same distribution. Tasks with suffix greater than 1 evaluate the OOD generalization capabilities of the model, and simulate biologically-relevant application scenarios (see Appendix C.1 for a detailed description of each task).

**ExtendedCHAMMI dataset.** We extend the CHAMMI train set to a total of $7,748,662$ images, incorporating additional data from both the source datasets and new data sources. The extended training dataset preserves the OOD characteristics of the CHAMMI tasks (see Appendix C.2).

**Meter-ML dataset.** The Meter-ML dataset contains images acquired by multiple sensors: four channels for NAIP (National Agriculture Imagery Program in the USA) images at resolution $720 \times 720$, four channels from Sentinel-2 at resolution $72 \times 72$, and lower resolution images from Sentinel-2 (S2) and Sentinel-1 (S1). The task consists in classifying sources of methane emissions in six categories (CAFOs, Landfills, Mines, ProcessingPlants, RefineriesAndTerminals, WWTreatment).

### 4.1.2 Implementation details

When pre-training the models, care was taken to ensure that the models processed the same amount of data. For the DINO BoC model, a sample consists of a single channel, whereas for the other models, a sample is an image with all of its channels. Therefore the former must be trained for more iterations to achieve fair comparison. For each pre-training dataset, the Channel-ViT and DINO HA models were pre-trained for $45,000$ iterations. Taking into account the average number of channels in the pre-training dataset, the Bag of Channels model was trained for a proportionally larger number of epochs. The batch size used was of 1024 for all models, and the batch size per GPU was set to 8, except for DINO BoC model, for which 32 fits in memory. On our largest dataset, we trained the models for about 2 days using 16 nodes, or 4 nodes for the DINO BoC model. Unless specified otherwise, we trained ViT large models. More details are provided in Appendix E. On the ExtendedCHAMMI dataset, the channel-adaptive models were pre-trained with balanced sampling across data sources.

### 4.2 Scaling channel agnostic feature representations with DINOv2

To assess the merits of channel-adaptive SSL strategies at scale, we assembled ExtendedCHAMMI as a large and diverse pre-training dataset, and compared the performance of channel-adaptive models on HPA-FOV, WTC, and JUMP-CP test-sets against fixed-channel baselines (Doron et al., 2023) pre-trained on ExtendedCHAMMI subsets with channels matching the respective target test sets (Table 1). The JUMP-CP dataset is not included in ExtendedCHAMMI; therefore it evaluates the generalization capability of channel-adaptive models on novel data sources. Results for an ablation removing datasets from the ExtendedCHAMMI dataset are presented in Appendix I, and results on all eight JUMP-CP channels are listed in Appendix J.

Surprisingly, we observed that models trained to independently encode channels (BoC), especially when trained with DINOv2, outperform joint-channel encoding models (Channel-ViT) across all tasks, and often with a substantial margin. DINO BoC also performs better than fixed-channel models on three out of four tasks, including the novel JUMP-CP dataset. Notwithstanding intuitions about the value of learning channel interactions in self-supervised pretraining, these results indicate that, at larger scale, BoC models are more effective in leveraging diverse microscopy data, and learn improved encoders compared to SOTA channel-adaptive methods that encode channels jointly. Although DINOv2 substantially outperforms MAE, our results suggest that BoC's advantages hold for other SSL objectives as well (Table 1). DINO HA performs on par with DINO BoC on this IID benchmarking setup. Next, we evaluated the impact of the network size on DINO BoC. Remarkably, while ViT-L expectedly leads to improved performance, even with a ViT-S, DINO BoC outperforms both ViT-L MAE BoC and Channel-ViT models.

Importantly, in several domains, MAE has been shown to yield features that are particularly well-suited to fine-tuning (He et al. (2022); Girdhar et al. (2023); Huang et al. (2022)). We hence also compare DINO BoC with MAE pretraining in a finetuned setting, which boosts the performance of all models on all datasets, excepted for Cell Types classification. Of particular relevance to practitioners, our results indicate that DINO BoCperformance advantage over MAE persists in fine-tuning regimes, with consistently better results. On the two most challenging tasks, namely protein localization, and JUMP-CP perturbation classification, the performance boost brought by DINOv2 is particularly strong (MAE-BoC: +2.5, DINO-BoC +4.7). Absent the ability of directly pre-training Microsnoop on ExtendedCHAMMI (training code is not available), the performance of MAE BoC serves as a proxy for Microsnoop performance, as it uses the same SSL method

employed in Table 2. In addition, we demonstrate in Table 2 that DINO BoC outperforms Microsnoop on the challenging Cyclops dataset (on which Microsnoop reported the largest gains) by a substantial margin, while especially fortifying performance on rare classes (up to 20 points). Unexpectedly, the BoC approach, and DINO BoC in particular, thus not only matches, but outperforms joint-channel-encoding, including on protein-localization prediction, setting a new state-of-the-art.

Table 1: **Comparison of channel-adaptive models trained on the ExtendedCHAMMI dataset and fixed-channel models.** The first three rows (CellDINO) are fixed-channel baseline models, separately pre-trained either on the HPA-FOV, JUMP-CP or WTC datasets. Best channel-adaptive results in bold; best results overall with frozen backbones are underlined. Note that the fixed-channel models can only be evaluated on data with matching number of channels.

| Model | SSL method | Network size | Channel adaptive | Training set | HPA-FOV F1 Protein loc. | HPA-FOV F1 Cell type | JUMP-CP Accuracy | WTC F1 Cell cycle st. |
|---|---|---|---|---|---|---|---|---|
| CellDINOv2 | DINOv2 | ViT-L | ✗ | HPA-FOV | 65.0 | 89.3 | ✗ | ✗ |
| CellDINOv2 | DINOv2 | ViT-L | ✗ | JUMP-CP | ✗ | ✗ | 44.3 | ✗ |
| CellDINOv1 | DINOv1 | ViT-L | ✗ | WTC | ✗ | ✗ | ✗ | 82.3 |
| Channel-ViT | DINOv2 | ViT-L | ✓ | ExtendedCHAMMI | 57.4 -7.6 | 90.4 +1.1 | 39.4 -4.9 | 87.2 +4.9 |
| DINO HA | DINOv2 | ViT-L | ✓ | ExtendedCHAMMI | 61.4 -3.6 | **91.3** +2.0 | **45.2** +0.9 | **91.0** +8.7 |
| BoC | MAE | ViT-L | ✓ | ExtendedCHAMMI | 54.0 -11.0 | 90.8 +1.5 | 39.3 -5.0 | 89.4 +7.1 |
| | DINOv2 | ViT-S | ✓ | | 55.6 -9.4 | 90.7 +1.4 | 44.5 +0.2 | **91.0** +8.7 |
| | DINOv2 | ViT-L | ✓ | | **61.7** -3.3 | 91.1 +1.8 | **45.2** +0.9 | 90.5 +8.2 |
| BoC fine-tuned | MAE | ViT-L | ✓ | ExtendedCHAMMI | 64.3 | 90.1 | 57.5 | 93.1 |
| | DINOv2 | ViT-L | ✓ | | **66.8** | **90.4** | **62.2** | **93.6** |

Table 2: **Comparison to Microsnoop on the Cyclops dataset.** DINO BoC dramatically outperforms Microsnoop, especially on the 4 least frequent classes (out of 16).

| | Class frequency | Budtip 1.5 % | Cell periphery 1.9% | Budneck 2.4% | Actin 3.8% | All |
|---|---|---|---|---|---|---|
| Microsnoop (Xun et al., 2024) | | 32.1 | 96.4 | 43.4 | 48.0 | 75.9 |
| DINO BoC | | **62.1** | **97.5** | **72.6** | **63.7** | **83.1** |

## 4.3 Cross-dataset generalization

Despite exhibiting superior scaling properties (at least for fluorescent microscopy data), it is not clear whether independent-channel encoding also provides benefits when confronted with unseen datasets and channel combinations - a key application for foundation models in this field. We hence compared cross-dataset generalization capabilities of DINO Channel-VIT, DINO HA , and DINO BoC models. To this end, we trained models exclusively on either on HPA-FOV or JUMP-CP, and evaluate them on HPA-FOV, JUMP-CP and WTC test sets. Once more, we find that DINO BoC outperforms DINO Channel-ViT on all cross-dataset tasks (Table 3). Noticeably, these results suggest that cross-channel reasoning is detrimental for the emergence of general, rather than specialized, dataset-specific features via SSL pretraining. DINO HA yields consistently better performances than Channel-ViT. Nevertheless, DINO HA still falls behind DINO BoC(Table 3), corroborating the conclusion that independent-channel-encoding is the winning strategy for channel-adaptive models.

Table 3: **Cross-dataset generalization of channel-adaptive models**. DINO BoC shows superior performance on unseen channel combinations. In-dataset results are shown in gray for reference.

| Model | Channel adaptive | Training set | HPA-FOV F1 Protein loc. | HPA-FOV F1 Cell type | JUMP-CP Accuracy | WTC F1 Cell cycle st. |
|---|---|---|---|---|---|---|
| DINO Channel-ViT | ✓ | HPA-FOV | 65.5 | 90.9 | 35.3 | 80.0 |
| DINO HA  (Ours) | ✓ | HPA-FOV | 66.7 | 91.3 | 37.3 | 88.9 |
| DINO BoC  (Ours) | ✓ | HPA-FOV | 65.2 | 91.5 | **40.2** | **89.8** |
| DINO Channel-ViT | ✓ | JUMP-CP | 29.5 | 82.0 | 53.4 | 81.8 |
| DINO HA  (Ours) | ✓ | JUMP-CP | 30.3 | **85.2** | 52.0 | 84.2 |
| DINO BoC  (Ours) | ✓ | JUMP-CP | **31.6** | 85.0 | 41.3 | **90.5** |

### 4.4 Out-of-distribution generalization on CHAMMI

We next assessed out-of-distribution generalization of channel-adaptive models leveraging the challenging CHAMMI benchmark set. Table 4 reports results for the 1-NN classifier protocol as defined by Chen et al. (2024). Given the modest size of CHAMMI's training set (ca. 100k images), the CHAMMI benchmark thus far is dominated by supervised channel-adaptive models, trained from scratch, that leverage labels for training. The top-performing model, HyperNet, inspired by Hypernetworks (Ha et al., 2017), uses a CNN backbone and an MLP that generates kernel weights for the initial convolutional layer of each input channel. Consistent with our previous results, we find that DINO BoC outperforms DINO Channel-ViT on OOD tasks for two out of three datasets. Remarkably, DINO BoC further exceeds the best supervised baseline on average OOD scores for WTC (5.3%) and CP (3, 8%). For completeness, we also report DINO HAresults, which underperform.

SSL methods tend to benefit from pre-training on a larger corpus of data. Consistently, we find that pretraining on our ExtendedCHAMMI dataset further improves the performance of channel-adaptive SSL models (Table 5) compared to the default CHAMMI training set (Table 4), even though part of the additional data comes from datasets unrelated to those on which the models are evaluated on.

Table 4: **F1 scores for 1-NN search on the CHAMMI test set. The models were pre-trained on the CHAMMI train split.** Lines 1-6 report the results of Chen et al. (2024) for CNN-based models trained from scratch in a *supervised* fashion. Line 1 reports the performance of FixedChannels, that consists on a separate model trained for each fixed channel combination. Lines 2-6 are channel-adaptive models. Lines 7-8 are self-supervised channel-adaptive ViTs. Best results between channel-adaptive self-supervised approaches are in bold.

| | Model | Average OOD | | | | WTC | | HPA | | | CP | | | |
|---|---|---|---|---|---|---|---|---|---|---|---|---|---|---|
| | | Mean | WTC | HPA | CP | Task1 | Task2 | Task1 | Task2 | Task3 | Task1 | Task2 | Task3 | Task4 |
| supervised | FixedChannels | 50.0 | 64.8 | 59.2 | 25.9 | 64.9 | 64.8 | 80.7 | 76.3 | 42.1 | 66.0 | 48.1 | 23.0 | 6.6 |
| | Depthwise | 51.7 | 65.2 | 64.4 | 25.6 | 68.9 | 65.2 | 84.9 | 81.3 | 47.5 | 67.3 | 47.8 | 22.4 | 6.5 |
| | TargetParam | 49.6 | 59.0 | 62.3 | 27.3 | 69.5 | 59.0 | 83.7 | 79.4 | 45.2 | 71.7 | 50.8 | 23.4 | 7.7 |
| | SliceParam | 45.7 | 56.8 | 54.6 | 25.6 | 61.6 | 56.8 | 77.0 | 69.0 | 40.3 | 64.6 | 47.5 | 22.2 | 7.1 |
| | HyperNet | 53.7 | 66.1 | 67.1 | 27.8 | 72.6 | 66.1 | 88.7 | 85.8 | 48.3 | 72.0 | 51.7 | 24.7 | 6.9 |
| | Template mixing | 46.6 | 56.5 | 57.7 | 25.7 | 63.1 | 56.5 | 80.8 | 74.1 | 41.3 | 67.1 | 46.8 | 22.7 | 7.5 |
| | DINO Channel-ViT | 42.6 | 45.3 | **53.6** | 29.0 | 68.7 | 45.3 | **92.2** | **65.2** | **42.0** | **95.2** | 51.5 | **25.2** | 10.3 |
| | DINO HA  (Ours) | 40.4 | 46.3 | 47.4 | 27.5 | 62.4 | 46.3 | 88.7 | 59.9 | 34.8 | 94.3 | **59.0** | 16.6 | 6.8 |
| | DINO BoC (Ours) | **48.8** | **71.4** | 43.3 | **31.6** | **79.4** | **71.4** | 87.0 | 56.4 | 30.2 | 93.5 | 58.5 | 20.1 | **16.3** |

#### 4.4.1 Linear probe for CHAMMI

We note that supervised methods are prone to encoding spurious correlations inherent to training sets, leading to poor performance on OOD tasks (Pernice et al., 2023; Mao et al., 2021; Barbu et al., 2019). By contrast,

Table 5: **F1 scores for 1-NN search on the CHAMMI test set. The self-supervised models were pre-trained on the ExtendedCHAMMI dataset.** Line 1 presents the best performing *supervised* baseline (HyperNet), which can only be trained on the annotated subset of CHAMMI.

| Model | Average OOD | | | | WTC | | HPA | | | CP | | | |
|---|---|---|---|---|---|---|---|---|---|---|---|---|---|
| | Mean | WTC | HPA | CP | Task 1 | Task 2 | Task 1 | Task 2 | Task 3 | Task 1 | Task 2 | Task 3 | Task 4 |
| HyperNet | 53.7 | 66.1 | 67.1 | 27.8 | 72.6 | 66.1 | 88.7 | 85.8 | 48.3 | 72.0 | 51.7 | 24.7 | 6.9 |
| DINO Channel-ViT | 43.6 | 46.2 | **55.6** | 28.9 | 64.5 | 46.2 | **92.1** | **65.3** | **45.9** | 89.0 | 53.5 | 21.8 | 11.3 |
| DINO HA  (Ours) | 43.2 | 54.3 | 46.9 | 28.4 | 66.4 | 54.3 | 90.1 | 60.4 | 33.5 | **93.6** | 57.1 | 15.8 | 12.1 |
| DINO BoC (Ours) | **51.6** | **79.0** | 43.0 | **32.7** | **79.4** | **79.0** | 86.6 | 59.3 | 29.6 | 92.6 | **57.6** | **22.1** | **18.5** |

our results in CHAMMI's 1-NN OOD regime indicate that self-supervision, and channel-adaptive SSL with BoC in particular, yields less biased representations that are substantially more robust to domain shifts. Still, while supervised methods that naturally promote clustering of the data in the embedding space according to their labels, SSL methods have been shown to yield nested embedding spaces (Doron et al., 2023). For example, for the HPA task, DINO BoC features first cluster by cell type, while the protein localization are retained as a nested factor of variation (see Appendix G). This organization is likely suboptimal for nearest-neighbor classification of e.g. novel cell type (HPA Task 2), or for a known cell type but novel protein localization categories (HPA Task 3). We hence further evaluated model performance using a linear probe.

Table 6: **F1 scores for a linear probe on CHAMMI test set.** Self-supervised models were pre-trained on the ExtendedCHAMMI dataset, while the supervised HyperNet was pre-trained on CHAMMI.

| Model | Average OOD | | | | WTC | | HPA | | | CP | | | |
|---|---|---|---|---|---|---|---|---|---|---|---|---|---|
| | Mean | WTC | HPA | CP | Task1 | Task2 | Task1 | Task2 | Task 3 | Task1 | Task2 | Task3 | Task4 |
| HyperNet | 65.4 | 85.3 | 82.9 | 27.9 | 87.9 | 85.3 | 94.4 | 92.5 | 73.2 | 93.5 | 51.5 | 17.3 | 15.0 |
| DINO Channel-ViT | 59.8 | 66.9 | **76.7** | 35.9 | 83.1 | 66.9 | 88.2 | **84.9** | **68.4** | 80.5 | 54.5 | 23.3 | 30.0 |
| DINO HA (Ours) | 62.7 | 76.2 | 72.4 | 39.5 | 88.0 | 76.2 | **88.5** | 82.4 | 62.4 | **91.7** | **61.6** | **27.5** | 29.3 |
| DINO BoC (Ours) | **67.9** | **89.2** | 74.9 | **39.7** | **90.5** | **89.2** | 88.3 | 84.7 | 65.0 | 90.5 | 60.5 | 25.8 | **32.7** |

Linear probing is also of particular interest for DINO BoC: unable to learn channel interactions explicitly, we note that DINO BoC , on 1-NN evaluation, underperforms both Channel-ViT and supervised baselines on the HPA protein localization tasks, which require spatial reasoning across channels. Even the minimal opportunity provided by linear probing to disentangle relevant from irrelevant information, and for DINO BoC  in particular, to relate information across channels, may thus further improve the performance of SSL models.

Consistently, a comparison of Tables 5 and 6 shows that the use of a linear probe substantially improves the scores of SSL channel-adaptive models overall, and of DINO BoC  on HPA Task 2 and 3 in particular. In result, DINO BoC  performs comparably to Channel-ViT, as well as DINO HA, on all protein localization tasks. In result, DINO BoC  surpasses all other models, including the HyperNet supervised baseline, on the mean OOD score, suggesting that ample information suitable to cross-channel integration is preserved in its channel-specific embeddings to be readily utilized post-hoc. This highlights the potential of this simple strategy to yield a powerful biological feature extractors for wide-ranging applications.

## 4.5   Experiments on geospatial imagery

We next asked whether the merits of DINO BoC  are limited to fluorescent microscopy applications, or whether the approach could be useful in other domains. We hence evaluated DINO BoC on the geospatial dataset Meter-ML, introduced by Zhu et al. (2022) (Table 7).

Zhu et al. (2022) showed that the highest resolution NAIP images lead to better performance, and that using all bands could improve the result of one class accuracy. We thus trained one DINO BoC model on NAIP, and one model on all all the S1, S2 and NAIP channels.

Table 7: **DINO BoC outperforms the state-of-the-art models when using all channels on the Meter-ML dataset.** Top: AUROC of models trained on NAIP and Sentinel data. Bottom: AUROC of models using only NAIP channels at inference.

| Approach | Architecture | Channel adaptive | Test dataset | Pre-training dataset | mAP |
|----------|--------------|------------------|--------------|----------------------|-----|
| Meter-ML Zhu et al. (2022) | DenseNet-121 | ✗ | NAIP, S2, S1 | ImageNet | 51.7 |
| LHRS-bot Muhtar et al. (2024) | VLM | ✗ | NAIP, S2, S1 | LHRS-Align-Recap (1.1M images & text) | 71.8 |
| VHM Pang et al. (2024) | VLM | ✗ | NAIP, S2, S1 | VersaD (14M images & text) | 72.7 |
| DINO Channel ViT | ViT-L | ✓ | NAIP, S1, S2 | NAIP, S1, S2 train (85K) | 64.4 |
| DINO BoC (ours) | ViT-L | ✓ | NAIP, S1, S2 | NAIP, S1, S2 train (85K) | **78.1** |
| Approach | Architecture | Channel adaptive | Test dataset | Pre-training dataset | mAP |
| Meter-ML Zhu et al. (2022) | DenseNet-121 | ✗ | NAIP | ImageNet | 54.8 |
| Supervised Cong et al. (2022) | ViT-L | ✗ | NAIP | fMoW Sentinel (770K) | 69.7 |
| SatMAE Cong et al. (2022) | ViT-L | ✗ | NAIP | fMoW Sentinel (770K) | 76.9 |
| ScaleMAE Reed et al. (2023) | ViT-L | ✗ | NAIP | fMoW RGB (363K) | 78.4 |
| USatMAE Irvin et al. (2023) | ViT-L | ✗ | NAIP | USAtlas NAIP (3.6M) | **83.7** |
| DINO Channel ViT | ViT-L | ✓ | NAIP | NAIP, S1, S2 (85K) | 63.8 |
| DINO Channel ViT | ViT-L | ✓ | NAIP | NAIP (85K) | 69.3 |
| DINO HA | ViT-L | ✓ | NAIP | NAIP, S1, S2 (85K) | 77.3 |
| DINO HA | ViT-L | ✓ | NAIP | NAIP (85K) | 77.6 |
| DINO BoC (ours) | ViT-L | ✓ | NAIP | NAIP, S1, S2 (85K) | 81.3 |
| DINO BoC (ours) | ViT-L | ✓ | NAIP | NAIP (85K) | **81.9** |

Using the exact same normalization and evaluation protocol as for the microscopy benchmarks, DINO BoC outperforms the state-of-the-art by a large margin when using all channels. Using only NAIP imagery, it is close to state-of-the-art approaches, even though it is pre-trained with much smaller datasets (two orders of magnitude less than the approach of Irvin et al. (2023)), and does not use remote sensing specific architectures. This strongly supports the generality of our approach to a wider range of imaging domains.

## 4.6 DINO BoC on ImageNet

Given major performance gains we observed for DINO BoC , we wondered whether the merits of independent channel-processing are limited to technical imaging domains in which (in contrast to RGB images) channels typically capture highly uncorrelated, distinct structures (Figure 2) (Bao et al., 2024). Therefore, we assess the impact of the DINO BoC vs. default DINOv2 (Oquab et al., 2023) on the classical ImageNet1K dataset (Deng et al., 2009). Top-1 classification accuracies are provided in Table 8.

Table 8: **Mean accuracy on the ImageNet 1k val set.** ViT-L models, linear evaluation.

| Model | Channel adaptive | Batch size × Channels | Top-1 Accuracy | |
|-------|------------------|------------------------|----------------|--|
| fixed channels | ✗ | $1024 \times 3 = 3072$ | ImageNet1K | 81.9 |
| fixed channels | ✗ | $3072 \times 3 = 9207$ | ImageNet1K | 83.1 |
| DINO HA | ✓ | $1024 \times 3 = 3072$ | ImageNet1K | 78.6 |
| DINO BoC(Ours) | ✓ | $3072 \times 1 = 3072$ | ImageNet1K | 82.8 |

Surprisingly, removing joint-channel embeddings as a prior seems to be beneficial even for extraordinarily well-explored domains like RGB images. With a fixed number of data seen during training (lines one and three, 81.9 vs 82.8), the comparison is favorable to DINO BoC. If we compare results given a constant amount of gradient steps, the accuracy of the fixed channels approach is only slightly higher than DINO BoC (0.3 points). Given the vast flexibility of the BoC approach in terms of applications, this small difference only underlines the potential of DINO BoC. For completeness, we also included a comparison to DINO HA with a lower batch size to make it fit into memory.

## 5 Conclusions

We report results on a large-scale study into self-supervised channel-adaptive training strategies, as a step towards general-purpose feature extractors for imaging domains with heterogeneous channel combinations, and in particular for fluorescent microscopy. Scaling the BoC approach using Vision Transformers and the state-of-the-art self-supervised DINOv2 method, DINO BoC outperforms models that rely on inter-channel reasoning, and positions it as the leading channel-adaptive approach. In addition to its strong performances, DINO BoC is notable for its simplicity, lacking any priors; whereas joint-encoding methods can *adapt* to variable channel numbers up to some maximum (see Appendix J), DINO BoC is channel-agnostic, rendering it suitable to arbitrary channel combinations. We also note that even the theoretical advantage of a uniform embedding space produced by joint-encoding methods (Bourriez et al., 2024) compared to BoC in practice remains unclear (see Appendix H). Instead, we show that DINO BoC substantially outperforms Channel-ViT in generalization to unseen channel combinations, and OOD tasks at test time. More broadly, our results call the utility of joint-channel-encoding as a prior for SSL pretraining into question. Indeed, we find that the DINO BoC approach achieves performance on par with state-of-the-art results for geospatial images, as well as RGB images, out-of-the-box. The code is available at `https://github.com/facebookresearch/dinov2/blob/main/docs/README_CHANNEL_ADAPTIVE_DINO.md`.

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

# A Detailed overview

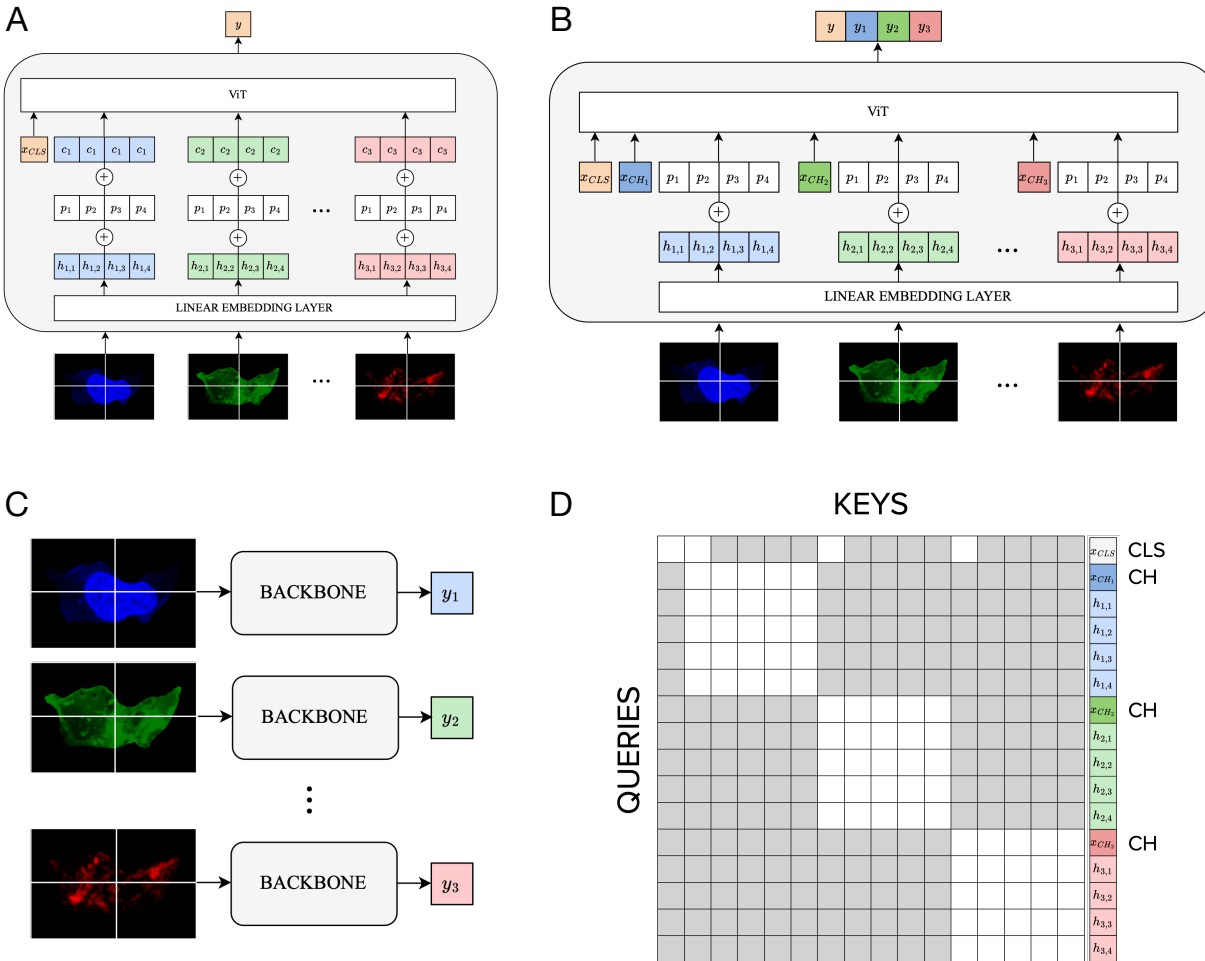

Figure 4: Overview of different channel-adaptive strategies. (A) Joint-channel-encoding strategy of Channel-ViT (Bao et al., 2024; Bourriez et al., 2024): the image is reshaped into a sequence of single-channel patches and channel embeddings are used to retain channel information. (B) Hierarchical Attention model: a specialized attention mask is used to enforce independent-channel-encoding via channel class tokens $x_{\mathrm{CH}}$, while the global class token $x_{\mathrm{CLS}}$ supports inter-channel reasoning. (C) Independent-channel-encoding strategy, note that a common backbone is used. (D) Attention mask of the Hierarchical Attention model.

# B Datasets

This section provides a detailed description of the datasets, and of the channels they encompass.

**Human Protein Atlas dataset.** The Human Protein Atlas (HPA) is an initiative that aims to map all human proteins across cells, tissues and organs. The subset of the data that is considered is the one of the Kaggle competition Human Protein Atlas (2019), concerned with the subcellular distribution of the proteins encoded by different genes. It covers 35 cell lines and 28 subcellular structures of protein localization. It covers thirty five cell lines and more than 13,000 proteins. The subcellular localization of each protein was classified into one or more of 28 subcellular structures.

The images were acquired using immunofluorescence and confocal microscopy. Four fluorescence dyes binding to (0) microtobules, (1) encoded protein, (3) nucleus and (4) endoplasmic reticulum are imaged in different channels. There are $113,545$ images in total.

There is also a single cell version of the same dataset Le et al. (2022) obtained through segmentation of the FOV images. The *HPA Single Cell* dataset contains $839,612$ images.

**WTC-11 dataset.** The dataset contains $214,037$ 3D images of cells, spanning 25 cellular structures. Other than tagging the structure of interest with a fluorescent protein (FP), fluorescent DNA and cell-membrane dyes were employed. The images have four-channels: bright-field; nucleus; cell membrane; structure of interest. Given that the focus of this work is to develop a foundation channel-invariant model for fluorescent microscopy, the bright-field channel was discarded.

The dataset provides cell-cycle stage annotations. The six possible labels are M0, M1M2, M3, M4M5, M6M7_single, M6M7_complete; where M0 through M7 denote cell cycle stages.

**Cell Painting dataset.** The Cell Painting Dataset Doron et al. (2023) used is the combination of the LINCS (Way et al., 2022), BBBC036 (Bray et al., 2017) and a third curated dataset (Moshkov et al., 2024), which includes BBBC022. All of those datasets were obtained following the Cell Painting protocol Bray et al. (2016), a standardized morphological profiling assay that images six fluorescent dyes in five channels, revealing eight cellular components. The components visualized in each channel are (0) nucleus; (1) endoplasmic reticulum; (2) nucleoli, cytoplasmic reticulum; (3) actin, golgi, plasma membrane; and (4) mitochondria. The goal of the studies included in the Cell Painting Dataset was to quantify the response of the cells to different perturbations: either compound treatments or gene over-expression experiments. Overall, the dataset includes 400 compounds and 80 gene over-expression experiments, totaling $8,423,455$ images.

**CHAMMI dataset.** The CHAMMI dataset was curated from the WTC-11, HPA Single Cell and Cell Painting datasets. It includes $65,103$ images from the WTC-11 dataset covering six tagged structures; $66,936$ images from the HPA Single-Cell dataset covering 18 cell lines and 8 protein localization classes, only images with a single protein localization annotation were selected; and $88,245$ images from the Cell Painting dataset covering seven compound experiments, including the negative control. In total there are $220,284$ images, of which $100,145$ are used for training.

**JUMP-CP dataset.** We considered the version of the JUMP-CP dataset used by Bao et al. (2024); it is a processed version of the data made available by the JUMP-Cell Painting Consortium Broad Institute (2021). Each image includes the five Cell Painting channels and three brightfield channels (HighZBF, LowZBF and brightfield).

The datasets generated by the JUMP-Cell Painting Consortium have the goal of enabling image-based drug mechanisms of action determination. As such, it encompasses multiple chemical and genetic perturbations. This particular version of the JUMP-CP dataset contains $229,228$ single cell images.

**OpenCell dataset.** The OpenCell dataset was introduced by Kobayashi et al. (2022), and consists of confocal images encompassing $1,311$ different tagged proteins. On average, each protein was imaged in 18.59 field of view images. Crops containing from 1 to 3 complete cells were extracted from each image, resulting in approximately 800 cropped images per protein. In total, $1,134,592$ images were made available. In addition to the tagged protein, a nuclear marker was used to visualize the nucleus. From the nuclear channel, they constructed a distance map and segmentation masks. However those two additional channels were not used for the purposes of this work.

## C   Details on the CHAMMI benchmark and dataset extension

### C.1   CHAMMI benchmark

The CHAMMI benchmark Chen et al. (2024) is a standardized evaluation framework for channel-invariant models. It presents a comprehensive set of nine tasks for channel-invariant models of varying complexity, that evaluates the ability of the models to generalize to new biologically-relevant experimental regimes. The images from each data source present in the CHAMMI dataset are split into one training set and several test sets, designed for specific tasks. Tasks with suffix 1 are IID classification problems, where the test and train data follow the same distribution. Originally, the CHAMMI benchmark considers a Nearest Neighbor (NN) evaluation.

The WTC-11 data is used for cell-cycle stage classification. The train set contains images with one of four cellular structures tagged: nuclear speckles, mitochondria, microtubules, or Golgi apparatus. Images of *W_Task2* are tagged with three novel cellular structures, the task evaluates whether the model is able to classify cell-cycle stages when an unseen cellular structure is tagged.

The HPA data supports protein localization classification. The train split covers 17 cell lines and four protein localizations: nuclear speckles, mitochondria, microtubules, or Golgi apparatus. The *H_Task2* images come from a novel cell line but covering the same protein localizations as the train split. The *H_Task3* images come from the same cell lines as the train split, but labeled with one of three novel protein localizations.

Lastly, the Cell Painting data is used for perturbation classification. The train set includes images of cells coming from 9 plates and perturbed with one of three treatments, as well as negative controls; The *C_Task2* images are perturbed with the same treatments as the train split and coming from the same data sources, however they belong to a set of 3 novel plates. The *C_Task3* includes the same treatments as the train split, but coming from the BBBC022 dataset and covering 4 novel plates. Finally, the *C_Task4* images are from the same set of plates and data sources as the train split, but the cells are perturbed with novel treatments.

For tasks that introduce new labels that are not present in the train set (HPA Task 3 and CP Task 4), a leave-one-out evaluation strategy is employed. Taking the example of HPA Task 3, the test data is split into sub-groups according to the cell line. Then, for each sub-group, the NN search is computed on both the training data and the remaining sub-groups. For CP Task 4, the data is split by plate ID.

### C.2   ExtendedCHAMMI dataset

The ExtendedCHAMMI dataset extends the CHAMMI train split to a total of $7,748,662$ images, using additional data from both the source datasets and new data sources, while preserving the OOD characteristics of the CHAMMI tasks.

In order to build the *ExtendedCHAMMI* dataset, the HPA FOV, HPA Single Cell, WTC-11, Cell Painting and OpenCell datasets were used. The samples belonging to the IID tasks W_Task1, H_Task1 and C_Task1 were removed from the WTC-11, HPA Single Cell and Cell Painting datasets, respectively. Furthermore, the images of the HPA FOV dataset containing cells present on H_Task1 were removed as well. With respect to the OOD tasks, the unseen tagged cellular structures for WTC-11; cell lines and protein localizations for HPA FOV and HPA Single Cell; and plates, data sources and treatments for Cell Painting were removed. The resulting number of images per dataset is summarized in Table 9.

## D   Hierarchical attention model training objective

The hierarchical attention model is based on a single-channel patch approach and, in addition to a global CLS token, it also inserts into the sequence channel CLS tokens. Moreover, it leverages a hierarchical attention mask (Figure 4), that constrains a channel's patch and CLS tokens to only interact with one another, while the global CLS token attends to the channel CLS tokens. Consequently, the model produces both a global image representation, and channel-level representations. In view of this, additional loss terms were included in the DINOv2 training objective, to account for the channel-level representations.

Table 9: **Data included on the ExtendedCHAMMI dataset.** The third column lists the amount of data that was included on ExtendedCHAMMI over the total size of the data source. For each data source, the number of channels and image type is specified: field-of-view (FOV) images, cropped FOV images containing a smaller number of cells, or images of a single cell. The last column lists which biological or experimental factors were discarded from the original datasets to preserve the OOD characteristic of the CHAMMI tasks.

| Dataset | Image type | Size | Channels | Discarded factors |
|---|---|---|---|---|
| **HPA Single Cell** | One cell | 296670/839612 | 4 | Cell line (HEK 293); protein localization (cytosol, endoplasmic reticulum, nucleoplasm) |
| **WTC** | One cell | 179994/214037 | 3 | Tagged cellular components (centrioles, tight junctions, actin bundles) |
| **OpenCell** | Cropped FOV | 1134592/1134592 | 2 | None |
| **Cell Painting** | Cropped FOV | 6103565/8423455 | 5 | Plates (SQ00015125, SQ00015168, SQ00015221); data source (BBBC022); treatments (BRD-K11129031, BRD-K62310379, BRD-K77947974) |
| **HPA FOV** | FOV | 33841/102190 | 4 | Cell line (HEK 293); protein localization (cytosol, endoplasmic reticulum, nucleoplasm) |

Consider an image $x$, and let $\mathcal{G}(x)$ denote the set of global crops of $x$, and $\mathcal{C}(x)$ the set of all crops of $x$, note that $\mathcal{G}(x) \subset \mathcal{C}(x)$. Furthermore, let $p_s^{[\text{CLS}]}(u)$ and $p_t^{[\text{CLS}]}(u)$ be the CLS tokens, transformed into probability vectors, output by the student and teacher networks for a crop $u$. Then, the DINO loss for a sample $x$ is:

$$\sum_{u \in \mathcal{G}(x)} \sum_{\substack{v \in \mathcal{C}(x) \\ v \neq u}} H\left(p_t^{[\text{CLS}]}(u), p_s^{[\text{CLS}]}(v)\right),$$

where $H(\cdot)$ denotes the cross-entropy. With the additional channel CLS tokens, $[\text{CH}_1], \ldots, [\text{CH}_C]$, the DINO loss is extended to:

$$\sum_{u \in \mathcal{G}(x)} \sum_{\substack{v \in \mathcal{C}(x) \\ v \neq u}} \left(\lambda_{\text{dino}}^{[\text{CLS}]} H\left(p_t^{[\text{CLS}]}(u), p_s^{[\text{CLS}]}(v)\right) + \lambda_{\text{dino}}^{[\text{CH}]} \sum_{i=1}^{C} H\left(p_t^{[\text{CH}_i]}(u), p_s^{[\text{CH}_i]}(v)\right)\right).$$

Another component of the DINOv2 loss is the KoLeo regularizer, that encourages the uniform span of features within a batch. Other than applying the KoLeo loss to the set of global CLS tokens for a batch, the loss is also separately applied to the set of all channel CLS tokens in the batch, with weights $\lambda_{\text{koleo}}^{[\text{CLS}]}$ and $\lambda_{\text{koleo}}^{[\text{CH}]}$. The remaining component of the DINOv2 loss, the iBOT masked-image-modeling loss, is left unchanged. We gave an equal weight to the losses on the global and channel CLS tokens. For the pre-training on the small scale CHAMMI dataset only, the DINO and KoLeo losses on channel class tokens were discarded, due to instabilities during training.

# E    Training and Evaluation details

## E.1    Pre-training details

Unless specified otherwise, we trained ViT-Large models with default patch size 16, and the default parameters of DINOv2 except a drop path rate of 0.1, a teacher momentum of 0.996, a learning rate of $5.0 \cdot 10^{-4}$, and 20 warm-up epochs. Transforms used include random contrast and brightness augmentations, flips and random resize crops of size 224.

## E.2    Evaluation details

The feature vector used in the evaluations – with the exception of the evaluations on the CHAMMI benchmark – are obtained by concatenating the CLS tokens across the last $L$ layers ($L = 4$), as well as the channel-wise

average pooled patch tokens. If $D$ is the dimension of the tokens (for a ViT-L $D = 1024$), and $K$ is the number of channels, then the feature size for Channel-ViT is $LD + KD$, for DINO BoC it is $LKD + KD$ and for DINO HA it is $L(1 + K)D + KD$.

For the evaluations on the CHAMMI benchmark (Tables 4, 5 and 6), only the CLS tokens are used and $L = 1$, therefore the feature size for Channel-ViT is $D$, for DINO BoC it is $KD$ and for DINO HA it is $(1 + K)D$. We ablate the impact of feature dimension in Appendix K.

We used the AdamW optimizer and a one cycle Cosine scheduler. We used the same train/val/test splits as Bao et al. (2024) for the JUMP-CP dataset. For HPA, we used the same train/val splits as Doron et al. (2023). For WTC, we created $80\% - 10\% - 10\%$ uniformly distributed train/val/test splits. For every evaluation, we trained 14 classifiers varying the learning rate between $10^{-4}$ and 1, and selected the best classifier on the val set. We trained all classifiers for 4350 iterations on 8 GPUs with a batch size per gpu of 32 for HPA-FOV and WTC, and 128 for JUMP-CP. To train the linear classifiers on HPA-FOV, the following transforms are used : random crop of size 384, flips and self normalization. For evaluation, a center crop of size $384 \times 384$ is taken, followed by self normalization. For JUMP-CP, we used the same normalization as in Bao et al. (2024) instead of self normalization, and crops of size 224. For WTC, we used self normalization and crops of size 224.

## F Influence of Hierarchical Channel Sampling

Bao et al. (2024) introduced a channel sampling technique for the training of ChannelViT, denoted hierarchical channel sampling (HCS). For an image $x$ with $K$ channels, HCS consists in sampling a number $m \in \{1, \dots, K\}$ uniformly at random and then randomly selecting $m$ channels from $x$ without replacement.

Results summarized in Table 10 suggest that Hierarchical Channel Sampling (Bao et al., 2024) hinders the performance of single-channel patch models. The channel sampling technique was found by Bao et al. (2024) to boost performance when pre-training and evaluating on the same dataset, mainly in the supervised scenario of missing channels at evaluation time. However it does not translate into improvements on the channel heterogeneous setting explored in this work. We postulate that it plays the role of a regularizer in the supervised context, but that this strategy is not well adapted to SSL.

Table 10: **Influence of HSC on DINO Channel-ViT models.** The models were pre-trained on the ExtendedCHAMMI dataset and evaluated on CHAMMI.

| Model | HCS | Average OOD | | | | WTC | | HPA | | | CP | | | |
| | | Mean | WTC | HPA | CP | Task1 | Task2 | Task1 | Task2 | Task3 | Task1 | Task2 | Task3 | Task4 |
|---|---|---|---|---|---|---|---|---|---|---|---|---|---|---|
| DINO Channel-ViT | ✗ | **43.6** | **46.2** | **55.6** | **28.9** | 64.5 | **46.2** | **92.1** | **65.3** | **45.9** | 89.0 | 53.5 | 21.8 | **11.3** |
| DINO Channel-ViT | ✓ | 39.9 | 39.5 | 51.9 | 28.4 | **66.4** | 39.5 | 88.5 | 62.3 | 41.5 | **90.2** | **56.0** | **22.3** | 6.9 |

## G Hierarchy of factors of variation in the feature space

SSL methods learn image representations using the samples themselves as the supervisory signal, therefore no label information controls the organization of the features in the embedding space.

In the channel-invariant models pre-trained on HPA FOV, we observe an emerging clustering of the features according to a hierarchy of semantic concepts, as illustrated in Figure 5. The features first cluster by cell type, while the protein localization is retained as a nested factor of variation.

## H Channels as confounders in a unified feature space

To assess the utility of a unified feature space produced by channel-invariant models such as the one proposed by Bourriez et al. (2024), we explore the effect of ablating channels in the HPA FOV dataset. Specifically, we remove nucleus and ER channels from a random half of the dataset and compare the resulting features against those of images with all channels within a jointly computed UMAP space (Figure 6).

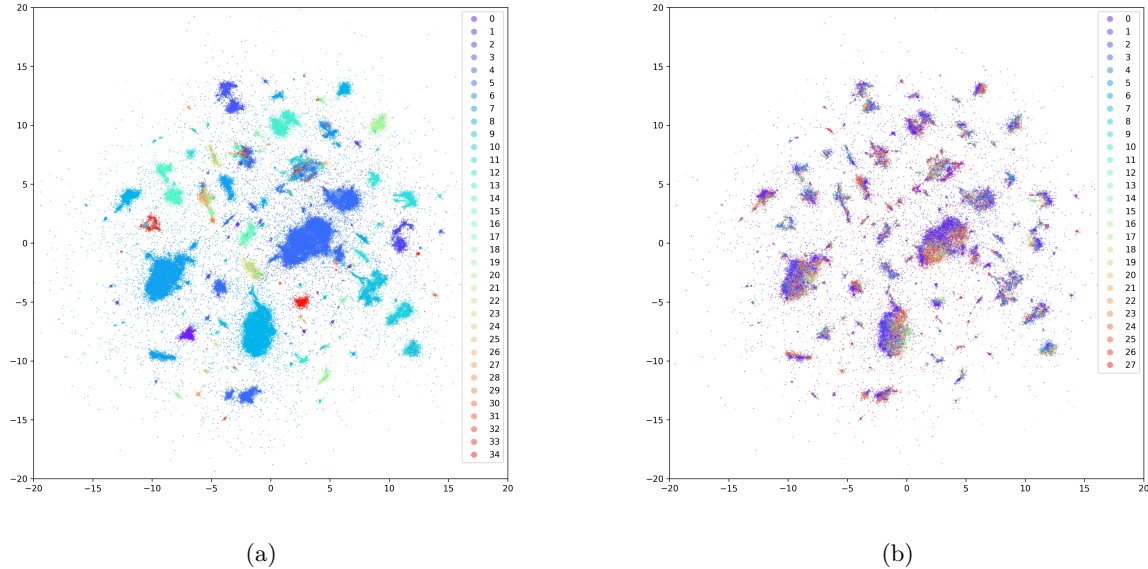

(a)                                                                (b)

Figure 5: **UMAP of the HPA FOV dataset highlighting different factors of variation.** UMAP space of the HPA FOV features obtained from the DINO BoC model pre-trained on the same dataset, colored according to (a) cell type and (b) protein localization. In (b) only samples with a single protein localization are displayed. The features are obtained separately for each channel, and then concatenated.

We observe that the data is clustered into distinct groups depending on the channels, and samples with different channels can hardly be compared to one another, even if their features have the same dimension. Therefore a common embedding space does not constitute an advantage of Channel-ViT over DINO BoC.

# I   Impact of the removal of different pre-training datasets

Table 11: **Ablation removing one dataset from the ExtendedCHAMMI dataset.** We report the linear evaluation results for DINO BoC.

| Training set | HPA-FOV F1 Protein loc. | HPA-FOV F1 Cell type | Accuracy on JUMP-CP | WTC F1 Cell cycle st. |
|---|---|---|---|---|
| ExtendedCHAMMI | 61.7 | 91.1 | 45.2 | 90.5 |
| minus WTC | 60.4 $_{-1.3}$ | **91.1** $_{-0.0}$ | 43.7 $_{-1.5}$ | 90.9 $_{+0.4}$ |
| minus Cell painting | 60.2 $_{-1.5}$ | **91.1** $_{-0.0}$ | 44.1 $_{-1.1}$ | 89.8 $_{-0.7}$ |
| minus HPA (FOV, single cell) | 41.7 $_{-20.0}$ | 89.9 $_{-1.2}$ | **46.9** $_{+1.7}$ | **92.3** $_{+1.8}$ |
| minus OpenCell | **60.9** $_{-0.2}$ | 91.0 $_{-0.1}$ | 44.2 $_{-1.0}$ | 90.0 $_{-0.5}$ |

To study the influence of specific pre-training datasets on the performance on others, we remove some pre-training datasets in Table 11 and evaluate the performance on HPA-FOV, JUMP-CP and WTC. As expected, when removing both HPA datasets, the protein localization performance is severely altered, but the cell type classification, a much easier task remains accurate. Not much difference is observed on the HPA tasks when removing the other datasets. In general, removing any dataset hurts the overall performance.

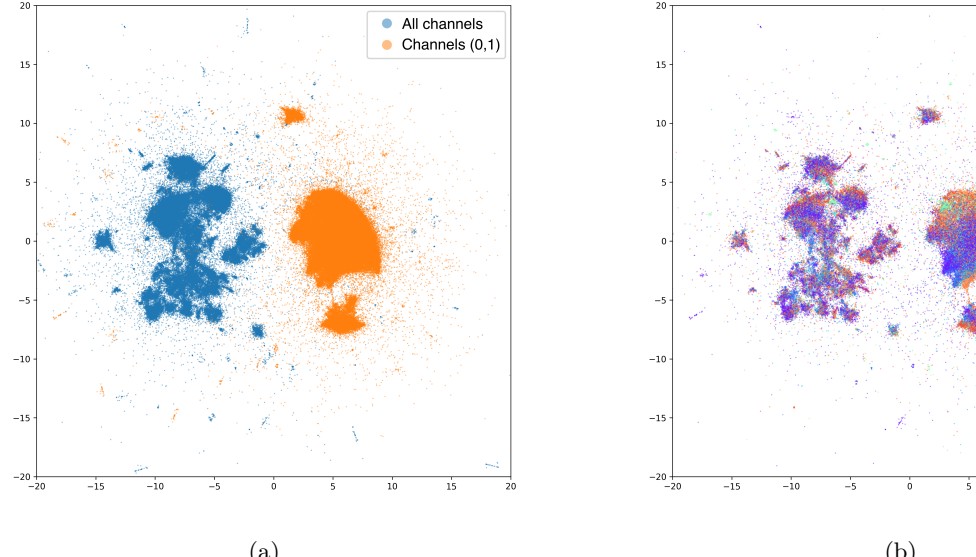

(a) (b)

Figure 6: **UMAP of the HPA FOV dataset to assess utility of unified feature space.** a) UMAP space of the HPA FOV features obtained from the Channel-ViT model pre-trained on the same dataset, comparing features computed when the model sees all four channels of the dataset, in blue, and features computed with only two channels (microtobules and protein), in orange. b) Same UMAP as in (a) but filtered for images with only one protein localization and colored according to them.

## J   Results on the eight channels of JUMP-CP

As shown in Table 12, both DINO HA and DINO BoC are flexible, resulting in improved performance when the downstream tasks involves a larger number of channels than at pre-training time. Here, the pre-training dataset contains up to 5 channels, while the models are evaluated with up to 8.

Note that the Channel ViT approach cannot be evaluated on images that contain more channels than the maximum number of channels per image of the pre-training data, since there are no trained channel embeddings for the extra channels.

Table 12: **Mean accuracy on the JUMP-CP dataset with models pre-trained on Extended-CHAMMI with a maximum of five channels.**

|  | JUMP-CP 5 channels | JUMP-CP 8 channels |
|---|:---:|:---:|
| Channel ViT | 39.5 | ✗ |
| DINO HA | **45.2** | 51.4 |
| DINO BoC | **45.2** | **51.6** |

## K   Ablation on feature dimension

In Appendix E.2 we describe how the features are obtained for each of the channel-invariant methods. In particular, joint and independent channel encoding strategies naturally lead to different sized embeddings, since the former results in an image level representation, while the later results in channel level representations.

Let $D$ denote the backbone's token dimension, $K$ the number of channels, and $L$ the number of last layers the CLS tokens are taken from. Then, when using both CLS and channel-wise average pooled patch tokens, the feature size for Channel-ViT is $LD + KD$, for DINO BoC it is $LKD + KD$ and for DINO HA it is $L(1 + K)D + KD$. When only CLS tokens are used, the feature size for Channel-ViT is $LD$, for DINO BoC it is $LKD$ and for DINO HA it is $L(1 + K)D$.

In order to demonstrate that DINO BoC outperforms DINO Channel-ViT due to the quality of the features and not due to its dimension, we consider two setups where both models have the same feature size. Those setups are:

1. For Channel-ViT we use only the CLS token from the last layer; while for DINO BoC we average pool the CLS tokens for each channel. Thus for both models the features are $D$-dimensional.

2. For Channel-ViT we concatenate the CLS token from the last layer to the channel-wise average pooled patch tokens, thus the feature is $D + KD$-dimensional. On the other hand, for DINO BoC we concatenate only the CLS tokens for each channel, resulting in a $KD$-dimensional feature.

The results obtained for those setups on the CHAMMI benchmark are listed in Table 13. In both cases DINO BoC outperforms DINO Channel-ViT.

Table 13: **F1 scores for a linear probe on CHAMMI test set: Ablation with similar embedding size.**

| | Feature | Average | OOD | | | WTC | | HPA | | | CP | | | |
|---|---|---|---|---|---|---|---|---|---|---|---|---|---|---|
| Model | dimension | Mean | WTC | HPA | CP | Task1 | Task2 | Task1 | Task2 | Task 3 | Task1 | Task2 | Task3 | Task4 |
| DINO Channel-ViT | $D$ | 59.8 | 66.9 | **76.7** | 35.9 | 83.1 | 66.9 | **88.2** | **84.9** | **68.4** | 80.5 | 54.5 | 23.3 | 30.0 |
| DINO BoC | $D$ | **65.4** | **86.7** | 67.9 | **41.5** | 89.4 | 86.7 | 82.9 | 79.1 | 56.7 | **83.8** | **61.2** | **26.6** | **36.8** |
| DINO Channel-ViT | $KD$ | 63.2 | 74.2 | **77.9** | 37.4 | 86.4 | 74.2 | **90.2** | **86.2** | **69.7** | 83.3 | 56.5 | 24.4 | 31.3 |
| DINO BoC | $KD$ | **67.9** | **89.2** | 74.9 | **39.7** | 90.5 | 89.2 | 88.3 | 84.7 | 65.0 | **90.5** | **60.5** | **25.8** | **32.7** |

## L Ablation on the training number of iterations

To compensate for a lower number of images seen during training, we trained BoC models for $K$ times more iterations than other models. We report in Table 14 the performance of the models when trained using the same number of iterations.

Table 14: **AUROC on the Meter-ML dataset in function of number of pretraining iterations**. Using NAIP images only.

| | Number of pretraining iterations | AUROC |
|---|---|---|
| Channel ViT | 34K | 69.3 |
| DINO HA | 34K | 77.3 |
| DINO BoC | 34K | 78.7 |
| DINO BoC | $4 \times 34K$ | 81.9 |

