# OpenReview forum: "Scaling Channel-Adaptive Self-Supervised Learning"
_TMLR — Accepted by TMLR_

### Review · Reviewer_R5tD · 2025-03-30

**Summary Of Contributions:**

The paper presents an extensive empirical study on self-supervised learning strategies for imaging domains with heterogeneous channel compositions. It introduces the DINO Bag of Channels (DINO BoC) approach, which independently encodes each channel using a shared Vision Transformer backbone and then aggregates the resulting embeddings, challenging the conventional idea that joint-channel encoding is always superior.

Additionally, the authors propose a novel Hierarchical Attention mechanism that limits inter-channel interactions to a high-level global token, exploring the balance between independent and joint encoding strategies. When scaled with the state-of-the-art DINOv2 framework, the DINO BoC method not only outperforms joint-channel encoding approaches across diverse benchmarks but also shows robust generalization to traditional RGB datasets like ImageNet 1k. Overall, it is a well-written manuscript, and I found their idea quite interesting with some concerns regarding their methodology.

**Audience:**

Yes

**Broader Impact Concerns:**

A limitation statement about the aggregation method used (only hierarchical attention, while there are many,) which can impact the result.

**Claims And Evidence:**

No

**Requested Changes:**

1. A separate figure for BOC and HA, and emphasize that what is the role of each component in the study. The current figure is somewhat hard to understand and differentiate between the work proposed and the role of each component.

2. Ablation study on the iterations of training since it is different from baseline. It can be one with the same iterations as the baseline and one with the current iterations to just show how the training time is impacting the performance of BoC.

3. authors' justification or empirical evidence with regard to w1 and w2 mentioned above.

**Strengths And Weaknesses:**

Strength: same as the contributions I mentioned above.

Weaknesses:
1. I have a question regarding the main claim in the paper: "our results call the utility of joint-channel-encoding as a prior for SSL pretraining into question." :
Different channels contain some information (tagged with the protein/stain etc.). My question is how the author ensures that the information within these channels is mutually exclusive (or fully independent) or not? If the channels carry information that does not have any added value for another channel, it is likely that joint-channel-encoding does not have any advantage over channel-invariant methods. So, I was pretty much looking forward to seeing an experiment that tries to answer this, but I haven't found anything.
Therefore, I suggest the authors design a controlled experiment where they pick specific channels (those with specific proteins tagged or sth similar) that are known to interact with each other and investigate their hypothesis again. It's likely that joint-channel-encoding methods perform well for those proteins, and underperform other methods in proteins with low interactions.

2. From the manuscript: "HA for testing the hypothesis that limiting inter-channel interactions enhances the performance of channel-invariant models". The best way to test this is using statistical hypothesis testing. Looking at both Tables 3 and 6, the results are mixed (compared to BoC); it seems inconclusive with regard to this hypothesis. One way to do it is to create different variations of the model (30 random seeds) and test on the datasets, and see if the authors can reliably test this with H-0 and H-1. Otherwise, it is better to adjust the claims in the paper.

---

> ### Author Response · Authors · 2025-04-15
> **Response to Reviewer R5tD**
>
> Thank you for your constructive feedback and the opportunity to improve our work. We hope that this revised version addresses your concerns:
> - **Overview Figures:** We have added a completely new diagrammatic overview figure (now Figure 2) that more clearly highlights key similarities and differences channel-adaptive vs. BoC-based channel-agnostic approaches pertaining to both practical, technical, and conceptual aspects. Along with an extensively revised Related Work and Method section, we think these changes substantially improve the clarity of presentation and the delineation of work from previous studies.
> - **Ablation Study:** We now include the suggested ablation using the same number of iteration for BoC than for Channel ViT in Appendix L. Note, however, that by decreasing the number of iterations for BoC to that of Channel-ViT models, BoC models see 1/C times less data, with C being the number of channels in a given dataset. Indeed, perfectly aligning training setups for Channel-ViT vs. BoC models is hard-to-impossible. For results reported in the paper, we kept batch-size (BS), learning rate (LR), and the amount of data seen constant, which in turn requires C-times more iterations for BoC models. To keep the number of iterations the same, one could train BoC models with C-times larger BS, but there is a clear yet difficult to control interaction between LR and batch-size, such that LR would need to be adjusted / tuned. We think our choice provides the fairest basis for comparison. Nevertheless, we intend to add the following ablation in the camera ready version:
>     - Train Channel-ViT 4x longer.
>     - Train DINO-BoC with 4x BS for the same number of iterations as Channel-ViT.
> - **W1 - Clarification on cross-channel reasoning:** A number of tasks we explore, and specifically the protein-localization tasks, require cross-channel reasoning, i.e. one would expect models that can learn across channels to be greatly advantaged. To provide further intuition on this: to determine whether or not a certain protein localizes to the Endoplasmic Reticulum or Microtubules (Tubulin), it would be exceedingly helpful to actually have reference information available on where these organelles localize within a given cellular image. That’s exactly the information that the other channels in the HPA dataset provide (see Fig. 1A). We show that, despite being unable to conduct any cross-channel computations, DINO-BoC not only succeeds at this task, but outperforms methods with explicit cross-channel reasoning. As we point out in paper, this suggests that DINO-BoC models retain substantial spatial information for a given channel in its output embeddings, which then allows downstream tasks to still complete tasks requiring cross-channel reasoning. A deeper analytical and theoretical exploration of this (admittedly very interesting) frontier is beyond the scope of this manuscript. Future research could center around perfectly-controllable synthetic datasets that directly assess cross-channel reasoning along with a structured investigation into what spatial information is (a) preserved and (b) indeed is necessary to solve certain tasks.
> - **W2 - Interpretation of DINO-HA results:** We understand that not including all HA results in most tables may have resulted in the misperception that BoC results are not consistently better. We now added HA results to Tables 1, 4, 5, 7, 8. Below, we report the differences as *HA score minus BoC score* (i.e. negative scores are in favor of DINO-BoC), with differences higher than 2 points in bold. This threshold is set arbitrarily, but it helps to visualize that the largest differences in performance are always in favor of DINO BoC:
>
>     - Table 1: -0.3, +0.2, 0.0, -0.5;
>     - Table 3: -1.3; +0.2,  **-2.9, -6.3;**
>     - Table 4: **-8.4;**
>     - Table 5: **-8.4;**
>     - Table 6: **-5.2** in average; detail: **-2.5, -13.0,** +0.2, **-2.3, -2.6,** +1.2 +1.1, +1.7, **-3.4**
>     - Table 7: **-4.3**
>     - Table 8: **-4.2**
>
>     We also agree that hypothesis-testing has a specific meaning that, in absence of formal testing (which would be very resource intensive), we do not meet. We therefore revised our sentence "... testing the hypothesis that limiting inter-channel interactions enhances the performance of channel-invariant models", replacing it with “This method can be seen as exploring a trade-off between Channel-ViT and DINO BoC, by limiting inter-channel interactions.”

---

> > ### Comment · Reviewer_R5tD · 2025-05-08
> > **Official Recommendation**
> >
> > I would like to thank the authors for their responses and revisions. Their replies have addressed the majority of my concerns, and the revised manuscript has improved in clarity and readability.
> > I still suggest that the authors consider adding a brief discussion on limitations or directions for future work, particularly with respect to W1, which remains an area worth highlighting.
> >
> > Having reviewed the other reviewers' comments and the corresponding authors' responses, I believe the paper’s contributions, especially its extensive experiments, are of interest to the community and outweigh its limitations.
> >
> > Therefore, my overall recommendation is **leaning accept**.

---

### Review · Reviewer_bUfp · 2025-04-01

**Summary Of Contributions:**

This manuscript proposes a channel-invariant self-supervised learning method, DINO BoC, for non-RGB scientific imaging. DINO BoC is a Bag of Channel (BoC) variant of DINOv2. The manuscript demonstrates that independently encoding channels consistently surpass joint-channel encoding strategies in extensive benchmarking experiments, including out-of-distribution and unseen channel combinations.

**Audience:**

Yes

**Broader Impact Concerns:**

None noted.

**Claims And Evidence:**

Yes

**Requested Changes:**

- Misuse of invariance terminology. Strictly speaking, invariance in representation learning implies that a model's representation remains unchanged under certain transformations or permutations. However, the authors use the term "channel-invariant" to describe models that can handle different numbers and types of input channels across datasets without retraining. Additionally, channels are not interchangeable, given that each imaging channel typically captures unique and complementary information, such as distinct cellular structures in microscopy or specific spectral bands in geospatial imagery.
- In multimodal learning, a common conceptual framework is to distinguish between inter-modality dependencies (relationships across different modalities) and intra-modality dependencies (relationships within a single modality). Recent frameworks, such as Inter- & Intra-Modality Modeling (Madaan et al., NeurIPS 2024), explicitly address these dependencies. The current submission explores analogous concepts but within the scope of multi-channel single-modality imaging data. Specifically, the DINO BoC method emphasizes intra-channel dependencies by independently encoding each channel without modeling explicit inter-channel interactions. Given this conceptual overlap, such multimodal frameworks could provide additional theoretical context and improve performance (as I noted in weaknesses about inconsistent performance on some tasks).

Madaan, Divyam, et al. "Jointly Modeling Inter-& Intra-Modality Dependencies for Multi-modal Learning." Advances in Neural Information Processing Systems 37 (2024): 116084-116105.

**Strengths And Weaknesses:**

Strengths:
- 7 datasets with fluorescent microscopy imaging.

Weaknesses:
- DINO BoC largely resembles a late fusion approach commonly used in multimodal learning, where each channel or modality is independently encoded before aggregating representations. From a methodological standpoint, the idea appears incremental.
- The overall performance boost appears to be inconsistent. It works better in some cases, while in others, it works worse.
- Only one dataset for geospatial imaging. It would be interesting to see whether claims shown with fluorescent microscopy imaging will hold.
- Tables for multiple experiments do not report results as mean$pm$std over multiple runs. Hence, the variability of the model's performance is not evident.

---

> ### Author Response · Authors · 2025-04-15
> **Response to Reviewer bUfp**
>
> Thank you so much for taking the time to review our paper! We hope to have addressed your concerns (and to have adequately integrated your suggestions) in our revision as summarized below:
> - **Incremental Idea:** We agree that combining DINOv2 with a Bag of Channel strategy is a simple idea. Moreover, as we discuss in the manuscript, there are a number of reasons to expect that more complex solutions (as exemplified by Channel-ViT) would be necessary for performant foundation models in channel-heterogenous domains to be trained at scale. The ChAda-ViT paper even provides some empirical evidence for that, albeit only at the hand of comparatively small datasets. Yet, with a first systematic investigation into the scaling-properties of channel-adaptive approaches, on the basis of a uniform set of modern architectures, SSL objectives, and a variety of challenging benchmarks, our study provides strong empirical evidence to refute this notion. Even though the combination of DINOv2 + ViTs + BoC has not been explored before and DINO-HA is new, the value of this work is not in its methodological innovations. Instead, the value of our work rests in the fact that this simple method (at possible risk of being abandoned by the field?) unexpectedly outperforms more complex methods when scaled, including in OOD generalization, a critical setting, which has remained largely unexplored in the channel-adaptive literature. Our results challenge previous theoretical assumptions and delivers a new  SOTA model, DINO-BoC, which, as a channel-agnostic model, is simultaneously substantially more flexible than contemporary joint-channel-encoding approaches. Therefore, we think our study will be of great interest to the community.
> - **Misuse of Terminology:** Thank you for pointing this out - we agree that “channel-invariant” is not ideal. We have removed “channel invariant” in favor of using “channel adaptive” as an umbrella-term, and distinguish “channel agnostic” approaches as a special case of channel-adaptive models that, in contrast to joint-channel-encoding approaches like Channel-ViT, can accommodate arbitrary channel numbers and combinations during training and at test time. We explicitly comment on this in our substantially expanded Related Work section, as well as our new diagrammatic figure (Fig. 2).
> - **Relation to multimodal literature:** We agree that there is ample ground for cross-fertilisation between the channel-adaptive and multi-modal learning literature, and we now discuss some analogies in the Related Work section (including the paper you have mentioned). The theoretical framework of “Inter- & Intra-Modality Modeling” (Madaan et al., NeurIPS 2024) cannot be applied off the shelf to our setup, being a supervised learning framework, but it certainly is a great reference for future work on a theoretical fundamentation for the experimentally observed superiority of independent-channel-encoding (which emphasizes intra-channel information) over joint-channel-encoding (which allows for inter-channel interactions). We added a paragraph in the related work section to discuss this, and also improved the presentation of our approach with a new diagrammatic figure (Fig. 2) that highlights parallels between the concepts of early vs. late fusion strategies and channel-adaptive methods.
> - **Variance of performance boost:** We agree that reporting stds across our experimental results would be ideal, and would aid the interpretation of what constitutes significant performance differences in each setting. However, computing these results across our extensive experiments would be very expensive. Similar to other studies, we therefore provide abundant reference scores to provide numerical context for each experimental setting. More importantly, we argue that the consistent and often quite substantial (e.g. Table-1) performance gain of DINO-BoC over Channel-ViT across all experiments leaves little room for doubt that our results are a statistical fluke. As we point out in the text, DINO-BoC outperforms (Tables 1&2), or performs at least comparably to Channel-ViT (OOD setting, Table-6) on HPA protein-localization task for which DINO-BoC (lacking cross-channel computation) would be expected to be particularly disadvantaged. Following suggestions from other reviewers, we have now also added DINO-HA results to all results tables which better illustrates a consistent performance gain of DINO-BoC over HA across our study (also see reply to R5tD). We plan to add a figure to summarize our results across all experiments for the camera-ready version.

---

### Review · Reviewer_jgAJ · 2025-04-02

**Summary Of Contributions:**

- A large scale comparative study of channel-invariant self-supervised learning methods.
- Shows that independent-encoding outperforms joint-encoding on in-domain, cross-dataset, and out-of-distribution experiments.
- Introduces a novel channel-invariant self-supervised learning method.
- Open sources said new SOTA general purpose feature extractor.

**Audience:**

Yes

**Broader Impact Concerns:**

No concerns.

**Claims And Evidence:**

Yes

**Requested Changes:**

### Related Work (Would strengthen)
- The paper mentions MAE, but there have since been derivatives better suited to training encoders on multiple channels/modalities. Addressing one such as MultiMAE https://arxiv.org/pdf/2204.01678 could strengthen the claim.

### Methodology (Critical)
- A textual explanation of DINO BoC needs to be included or be better signposted.
- The delimitation between the contributions of the paper and existing methods needs to be clearer.
- Describe only methods in Section 3 that are used for experimentation as it is unclear if which and if all methods are used. A brief overview of existing methods may be better suited in the related work section.

### Results (Would strengthen)
- The results for DINO HA are not far behind and sometimes superior to DINO BoC which weakens the claim that independent channels encoding is superior. Including results for DINO HA where possible in Tables 2, 4, 5, 7, and 8 would strengthen this claim.

**Strengths And Weaknesses:**

### Strengths
- Provides experimental results on a wide range of datasets in multiple domains.
- Detailed ablation studies and supplementary experiments.
- Source code will be open sourced.
- Includes brief details about unsuccessful routes of investigation.

### Weaknesses
- The methodology is unclear and it is difficult to differentiate between existing methods and the proposed methods. It appears the explanation of the claimed SOTA methodology, DINO BoC, is missing or is not clearly identified.
- Many results for DINO HA are close to and occasionally superior to DINO BoC. This weakens the evidence for the claim that the independent encoding of channels outperforms joint encoding.

---

> ### Author Response · Authors · 2025-04-15
> **Response to Reviewer jgAJ**
>
> Thank you for your review and great pointers. We have followed your suggestions for revision as outlined below:
>
> - **Related work - MAE:** Thank you for pointing out the Multi-MAE work that we now discuss in the related work. We also expanded our comparisons to MAE by adding fine-tunning results for both MAE-BoC and DINO-BoC to Table 1, since (as pointed out by R1 (i3CN)) MAE-based features have been shown to be particularly suitable to fine-tuning in several domains (imaging, video, audio). However, we find that DINO-BoC models continue to outperform MAE, including on the two most challenging tasks, namely protein localization, and JUMP-CP perturbation classification, with a boost of +4.7 for DINO-BoC vs. +2.5 for MAE-BoC.
> - **Methodology - Improved textual explanation of DINO-BoC:** We have significantly revised the Method section to more clearly identify technical details pertaining to each method we implement for our experiments (Channel-ViT baselines, DINO-BoC, and Hierarchical Attention). We have further added a new overview figure (Fig. 2) to clarify the approach at training and inference time, as well as highlighting differences with other channel adaptive strategies.
> - **Methodology - Revised Related Work Section:** We substantially rewritten and expanded the related work section to clearly delineate our contributions from previous work in each relevant field.
> - **Methodology - Restructured Method Section:** We have removed technical descriptions on methods (e.g. the original ViT) as to limit the Method section to only those details directly relevant to our implementations and experiments. We think keeping technical details pertaining to our Channel-ViT baselines is important, since (1) we re-implement an approach that has been proposed / used by at least three different studies, and (2) as it helps delineate BoC models more clearly.
> - **Results - DINO-HA:** We included DINO HA results in Tables 1, 4, 5, 7, 8. On the “easy” HPA cell and WTC tasks, with more than 90% accuracy, DINO HA performs slightly better than BoC. On the more challenging HPA protein and JUMPCP tasks, the two approaches perform on par.  In the vast majority of tasks (on Imagenet, Meter-ML, CHAMMI out of distribution different evaluations), DINO BoC significantly outperforms DINO HA (please see our response to reviewer R5tD for even more details).

---

### Review · Reviewer_i3CN · 2025-04-04

**Summary Of Contributions:**

The paper investigates the importance of a particular design choice (how channels are ‘fused’) when training foundation models to be applied in regimes where observations are image-like but may have diverse or missing channels across downstream datasets. The author’s find that a simple baseline: ensembling/concatenating the outputs of a model trained independently on individual channels/measurement types to be surprisingly effective, outperforming various other strategies for handling variable channel input on a variety of evaluations.

**Audience:**

Yes

**Claims And Evidence:**

Yes

**Requested Changes:**

- If at all possible, including some fine tuning experiments would improve the work in my eyes. If not possible I still think including some discussion of how results may change in this setting is warranted.
- I think some discussion of the relationship between the methods compared here and the broader multimodal learning setting al la Liang et al. 2024 would be good to have.
- Please include a more explicit description of the DINO BoC method (potentially as its own section).
DINO HA is missing from some (most?) results tables, I think including it for all evaluations would make the messaging even clearer.

**Strengths And Weaknesses:**

Strengths:
- The proposed Bag-of-Channels (BoC) approach is conceptually simple and performant.
- Extensive experiments are conducted on a wide variety of downstream tasks and two relevant domains (microscopy and geospatial imaging).
- Having the hierarchical attention model as I kind of “bridge” between the two extreme approaches is a nice touch.

Weaknesses:
- Lack of fine-tuning results: in many practical situations practitioners have a reasonable amount of data to conduct fine tuning on a foundation model for a specific task. Thus it would be interesting to determine whether there are advantages or disadvantages to the BoC approach to pretraining when fine tuning. Furthermore, it is well known that MAE style pretraining requires fine tuning to be competitive with contrastive pre training (see He et al. 2023 Masked Autoencoders are Scalable Vision Learners), so it would be more fair to compare DINO BoC to MAEs in this setting.
- I think there is an opportunity to connect to the broader literature on multi-modal learning that has been missed. For example Liang et al. 2024 “Factorized Contrastive Learning: Going Beyond Multi-view Redundancy” explores the importance of considering information that is unique to each modality (channel in this setting) along side the shared information. Could this framework be used to understand the success of BoC (i.e. channel mixing approaches squeeze out task relevant information that is not shared across channels/modalities)?
- The description of the BoC approach is somewhat difficult to understand given the presentation. My understanding is that it involves training one backbone on only single channel inputs (from each modality), and then at test time ensembling the outputs for each channel. I think stating this plainly in its own section (i.e. between 3.2 and 3.2.1) could improve the readability of the paper.

---

> ### Author Response · Authors · 2025-04-15
> **Response to Reviewer i3CN**
>
> Thank you for your review and great suggestions. We hope we address your concerns in this revision. Specifically:
> - **Fine-tuning:** We now report fine-tuning experiments in Table 1, comparing DINO BoC with MAE BoC. The  consistently higher performance of DINO BoC on the four tested tasks in this setting too lends further support to our conclusions, and shows that DINO BoC’s advantage over MAE is likely to persist as practitioners fine-tune the models. On the two most challenging tasks, namely protein localization, and JUMP-CP perturbation classification, the boost brought by DINOv2 is particularly strong (MAE BoC: +2.5, DINO-BoC: +4.7).
> - **Relation to multi-modal learning:** We thank you for bringing the “Factorized Contrastive Learning: Going Beyond Multi-view Redundancy” paper to our attention. Indeed there is a relevant connection between channel-adaptive learning and the multi-modal learning literature. We now discuss common aspects with multi-modal works in the Related Work section. Regarding the “Factorized Contrastive Learning: Going Beyond Multi-view Redundancy” paper, their framework and discussion of multi-view redundancy and task-relevant unique information cannot be directly translated to our scenario. This is due to the fact that in applying contrastive learning to multi-modal data, the different views come from the different modalities. As such the representations are trained to capture shared/mutual information, possibly at the expense of task-relevant information unique to each modality. This would be relevant in our case for joint-channel-encoding models (i.e. our Channel-ViT baselines), if we sampled views that contained different channels, and thereby (effectively) further emphasized cross-channel computation. However, in our case, all views of an image contain all of the channels (or a single one for BoC models). While therefore not directly comparable, we nevertheless note that both FactorCL and our study seem to point in the same direction: both argue for the importance of dedicating computation to individual modalities / channels.
> - **Improved description of DINO-BoC:** We revised the Method section such that methodological details pertaining to DINO BoC, as well as our Channel-ViT baseline and HA models are more explicitly identifiable. We have also replaced the accompanying diagrammatic figure with a new, higher-level design, that provides a (in our mind) much improved overview on the key conceptual and technical differences between current channel-adaptive methods. We also significantly revised the Related Work section to add further context and more clearly delineate our work from previous studies.
> - **Additional DINO-HA results:** We now include DINO HA results in Tables 1, 4, 5, 7, 8. On the “easy” HPA cell and WTC tasks, with more than 90% accuracy, DINO HA performs slightly better than BoC. On the more challenging HPA protein and JUMPCP tasks, the two approaches perform on par.  In the vast majority of tasks (on Imagenet, Meter-ML, CHAMMI out of distribution different evaluations), DINO BoC significantly outperforms DINO HA (please also see our response to reviewer R5tD).

---

### Comment · Editors_In_Chief · 2025-06-24

Post-publication, the authors submitted a revised PDF with the correct link for the code, which replaced the previous camera-ready paper on June 24.

---

### Decision · Action_Editor_8rVS · 2025-05-11

**Recommendation:** Accept as is

**Comment:**

The paper introduces a straightforward but empirically effective approach, called DINO BoC, for self-supervised learning in the context of scientific imaging with heterogeneous channels. It provides an extensive empirical evaluation, open-source resources, and suggests a shift in how practitioners might approach channel variation during pre-training.

However, the reviewers consistently pointed out the some weaknesses, which have claimed to be solved by the authors and were accepted by the reviewers.  These weaknesses are:

- Insufficient methodological clarity: The description of the DINO BoC architecture and its operational details should be isolated in a dedicated section for better comprehension. Claimed to be addressed.
- Incomplete evaluation: Fine-tuning results are absent, and the comparison with more recent multimodal or masked modeling methods (e.g., MultiMAE, Inter-/Intra-modality frameworks) is underdeveloped. Claimed to be addressed.
- Overly strong claims: Statements such as "independent encoding is better at scale" could be tempered given that DINO HA shows comparable performance in many settings and that variance across datasets is not quantified.
- Misuse of terminology: Terms like "invariant" may mislead, since the method handles channel heterogeneity, not invariance in a strict sense. Claimed to be addressed.

**Audience:**

Yes. TMLR’s audience, which includes researchers and practitioners interested in self-supervised learning, foundation models, scientific imaging, and multimodal learning, would find this paper relevant. The growing use of SSL in non-RGB domains such as microscopy and remote sensing means that effective channel-robust pre-training methods are of interest. The DINO BoC method, while simple, offers a practical and scalable alternative to joint encoding, making it attractive to those dealing with heterogeneous or sparse channel data.

**Claims And Evidence:**

Yes, the central claims of the paper, i.e., independently encoding channels using the proposed DINO BoC framework can outperform joint-channel encoding methods in multi-channel scientific imaging tasks, are supported by extensive experimental evidence. The authors benchmark across diverse domains (microscopy and geospatial imagery) and test both in-domain and out-of-distribution settings. However, several reviewers have raised concerns that may weaken the strength of the claims:
1) Methodology clarity: The DINO BoC method is not described with sufficient clarity, especially for readers unfamiliar with the variations being compared. The explanation of what exactly constitutes "independent encoding" could be clearer and placed more prominently. This issue has been addressed in the revised paper.
2) Mixed results: In some benchmarks, joint-encoding (e.g., DINO HA) performs comparably or better than BoC, raising concerns about the generality of the performance advantage. Statistical significance testing or performance variance (e.g., means ± stds) across multiple runs is missing, which would help substantiate the robustness of the claimed advantages.
3) Lack of fine-tuning results: The absence of fine-tuning experiments makes it hard to evaluate the downstream practical impact of BoC compared to methods like MAE, which are known to benefit significantly from fine-tuning. This issue has been addressed in the revised paper.
Even with the limitations above, the empirical evidence presented is enough to support the the paper’s claims.